# NME4 mediates metabolic reprogramming and promotes nonalcoholic fatty liver disease progression

Shaofang Xie[1,2,3], Lei Yuan[1,2,3], Yue Sui [ID][1,2,3], Shan Feng[3], Hengle Li[4] & Xu Li [ID][1,2,3]✉

## Abstract

**Nonalcoholic fatty liver disease (NAFLD) is mainly characterized by excessive fat accumulation in the liver, and it is associated with liver-related complications and adverse systemic diseases. NAFLD has become the most prevalent liver disease; however, effective therapeutic agents for NAFLD are still lacking. We combined clinical data with proteomics and metabolomics data, and found that the mitochondrial nucleoside diphosphate kinase NME4 plays a central role in mitochondrial lipid metabolism. Nme4 is markedly upregulated in mice fed with high-fat diet, and its expression is positively correlated with the level of steatosis. Hepatic deletion of Nme4 suppresses the progression of hepatic steatosis. Further studies demonstrated that NME4 interacts with several key enzymes in coenzyme A (CoA) metabolism and increases the level of acetyl-CoA and malonyl-CoA, which are the major lipid components of the liver in NAFLD. Increased level of acetyl-CoA and malonyl-CoA lead to increased triglyceride levels and lipid accumulation in the liver. Taken together, these findings reveal that NME4 is a critical regulator of NAFLD progression and a potential therapeutic target for NAFLD.**

**Keywords** NME4; Coenzyme A Metabolism; Lipid Accumulation; NAFLD
**Subject Categories** Metabolism; Molecular Biology of Disease

## Introduction

In recent years, with the changes that have occurred in people's lifestyles and diets, nonalcoholic fatty liver disease (NAFLD) has gradually become the most common chronic liver disease (Diehl and Day, 2017; Younossi et al, 2016); NAFLD affects approximately 30% of adults in the general population and up to 70% of patients with type 2 diabetes mellitus (Younossi et al, 2019; Younossi et al, 2016). NAFLD progresses from simple steatosis to nonalcoholic steatohepatitis, liver fibrosis, and even more severe stages, such as cirrhosis and hepatocellular carcinoma, ultimately leading to liver failure (Diehl and Day, 2017). Due to the complexity of the pathogenesis of NAFLD and the limited understanding of its

pathophysiology, there is still a lack of effective strategies for clinical intervention (Wong et al, 2016). Therefore, there is an urgent need to identify new therapeutic targets for NAFLD.

NAFLD is usually associated with the dysregulation of metabolic pathways, resulting in insulin resistance, steatosis, oxidative stress, and lipotoxicity (Muoio and Newgard, 2008). Recent studies suggest that the continuous adaptation or "remodeling" of mitochondrial energetics, gene expression, morphology, and content plays a key role in the pathogenesis of NAFLD/nonalcoholic steatohepatitis (NASH) (Patterson et al, 2016; Satapati et al, 2012). The dysregulation of mitochondrial energetics in NAFLD leads to an imbalance between lipid accumulation (uptake, synthesis) and disposal (secretion, oxidation) (Diraison et al, 2003) and hepatic steatosis. The accumulation of toxic lipid intermediates triggers inflammation and impairs insulin signaling (Maceyka and Spiegel, 2014; Patterson et al, 2016; Perry et al, 2014), leading to a series of chain reactions, including higher lipid peroxidation rates, the formation of cytotoxic aldehydes and the production of proinflammatory cytokines, and ultimately to cell death.

However, the exact mechanism and the key mitochondrial factors that are responsible for this process remain largely elusive. To obtain a better understanding of the initiation and progression of NAFLD and to identify key regulators in the mitochondria, we combined clinical dataset and multi-omics datas, and identified the mitochondrial nucleoside diphosphate kinase NME4 may play a central role in mitochondrial lipid metabolism.

NME4, which is also known as NDPK-M or NM23-H4, is a nucleoside diphosphate kinase that belongs to the nonmetastatic 23 (NM23) family (Herbert et al, 1955). Human NME4 is widely expressed at high levels in the liver; at intermediate levels in the heart and colon; and at low levels in the brain, testis, and peripheral leukocytes (Lacombe et al, 2009, 2018; Milon et al, 1997). NME4 specifically localizes to the mitochondrial intermembrane space via a specific N-terminal sequence, which must be cleaved and removed to allow its catalytic activity. NME4 binds to the inner mitochondrial membrane via anionic phospholipids, such as cardiolipin (Milon et al, 2000; Tokarska-Schlattner et al, 2008). The key functions of NME4 in mitochondrial physiology include (1) the transfer of oxidatively generated NTP to different nucleoside diphosphates, mainly the phosphorus of GDP, to generate GTP to provide local fuel, such as for mitochondrial GTPases, and (2) to promote the intermembrane transfer of cardiolipin in the mitochondria of living cells, which serves as a pro-mitochondrial

[1]Westlake Institute for Advanced Study, Fudan University, 310018 Shanghai, China. [2]Key Laboratory of Structural Biology of Zhejiang Province, School of Life Sciences, Westlake University, 310024 Hangzhou, Zhejiang, China. [3]Westlake Laboratory of Life Sciences and Biomedicine, 310024 Hangzhou, Zhejiang, China. [4]School of Life Sciences, Southern University of Science and Technology, 518055 Shenzhen, China. ✉E-mail: lixu@westlake.edu.cn

or pro-apoptotic signal (Schlattner et al, 2013; Tokarska-Schlattner et al, 2008). Given the importance of mitochondrial energy metabolism in NAFLD, we further investigated the functions of NME4 in NAFLD.

# Results

## NME4 expression is upregulated in fatty liver and correlates with NAFLD progression

To identify the key regulators of NAFLD in mitochondria, we analyzed NAFLD data in a comparative toxicogenomics database (Davis et al, 2021) and identified 2750 genes that are associated with NAFLD (Fig. 1A). We also analyzed the expression profile of the mitochondrial protein compendium (Pagliarini et al, 2008), and identified 1097 genes which encode proteins that are mainly localized to the mitochondria. Using a tissue-based map of the human proteome, we found 2389 genes are highly expressed in liver tissue (Uhlen et al, 2015). Eighty proteins were identified in all the three datasets, and these proteins are potentially involved in NAFLD (Fig. 1B). The expression level of the 80 candidate genes was presented in a line chart and heatmap, illustrating their differential expression in liver tissue (Figs. 1C and EV1A). Among these proteins, half was reported to be associated with NAFLD, including HMGCS2, ACAT2, SHMT1 (Asif et al, 2022; Liu et al, 2021; Mardinoglu et al, 2014; Romeo, 2022). We also identified NME4, which is a mitochondrial protein that maintains the balance between NTP and NDP, as a hit in this list.

To understand the detailed mechanism underlying NAFLD progression, a NAFLD mouse model was established by feeding mice a diet high in fructose, cholesterol, and fat content (HFD). Compared to the mice fed a normal diet, the HFD-fed mice showed severe hepatic steatosis, as evidenced by the results of hematoxylin and eosin (H&E) staining and Oil Red O staining (Figs. 1D–F and EV1B,C). Nme4 mRNA and protein levels were both significantly increased in the livers of the HFD-fed mice in a time-dependent manner (Figs. 1G–J and EV1D–F) and positively correlated with the level of steatosis (Fig. 1K). Although crosstalk between adipose and liver is an important event in the development of NAFLD, NME4 showed no significant difference in the adipose tissues of HFD mice fed with different time points (Fig. EV1G–J; Appendix Fig. S1A), indicating NME4 mainly functions in liver cells. These results suggest the specific role of NME4 in the development of liver steatosis. Studies have shown that Nme4 upregulation is associated with enhanced mitochondrial function in mice with HFD (Koliaki et al, 2015). To rule out the possibility that the upregulation of Nme4 only reflects a mitochondrial stress response, we examined the expression levels of mitochondria-related genes, including Ppargc1a, Ndufs7, Cox5a, and Cox8b, in liver tissue from 12- or 24-week normal and HFD-fed mice. The expression of genes associated with mitochondrial function was slightly downregulated in both the 12 and 24-week HFD-fed groups of mice, indicating the general mitochondrial function was not increased in our model (Appendix Fig. S1B,C).

Liver tissues contain several types of cells, regulating normal liver functions together. To further investigate the expression pattern of Nme4 in liver tissue, we search the single-cell database (Tabula Muris et al, 2018) and found Nme4 mainly

expressed in hepatocytes in 3-month mice liver tissue (Appendix Fig. S2A–C). In addition, in vitro experiments demonstrated that Nme4 expression increased in a concentration- and time-dependent manner in primary mouse hepatocytes treated with PO (palmitic acid/oleic acid = 1:2) (Fig. 1L,M). Together, these results suggested that NME4 may play an important role in NAFLD progression.

## Tandem mass tag-based quantitative proteomics revealed the functions of NME4 in lipid metabolism

To further elucidate the biological roles of NME4, we generated an NME4 knockout cell line using the CRISPR-Cas9 system (Fig. EV2A,B) and performed tandem mass tag (TMT) 6plex-based quantitative proteomics to study the changes in the cellular protein levels induced by NME4 depletion (Fig. 2A). We identified and quantified the levels of 8187 proteins (see Dataset EV1), among which 270 proteins were significantly upregulated and 22 proteins were significantly downregulated upon NME4 depletion (Figs. 2B and EV2C).

To provide a comprehensive overview of the cellular functions and diseases in which NME4 is involved, we performed gene ontology analyses using the proteins whose expression was significantly changed after NME4 depletion. As expected, significant changed proteins were highly enriched in pathways related to the cell cycle, cytoskeleton organization, and inflammation, as reported previously (Choudhuri et al, 2010; Ernst et al, 2021; Fuhs and Hunter, 2017a; Kar et al, 2012) (Fig. 2C). In addition to the reported functional categories, several other biological processes, including lipid metabolism and lipid transport, were also highly enriched (Fig. 2C). Proteomap analysis showed that lipid and steroid metabolism were highly enriched when NME4 was depleted, with proteins that are involved in these processes were highlighted (Figs. 2D,E and EV2D). Abnormal lipid metabolism has been associated with various metabolic diseases, such as obesity (Wang et al, 2014b), type 2 diabetes mellitus (Palomer et al, 2013), and NAFLD (Shen et al, 2014). These results indicate that NME4 is involved in NAFLD by regulating lipid metabolism.

## Nme4 is essential for lipid accumulation in HFD mice liver and in human liver cells

To evaluate the role of NME4 in NAFLD progression and lipid accumulation, we first generated an Nme4 knocked down mice by injecting AAV type 8 virus expressing Nme4 shRNA (AAV-sh*Nme4*) via tail vein, to primarily knock down Nme4 in mice liver, and fed mice with HFD for 12 weeks (Fig. 3A). Most of the viruses attach to the liver rather than adipose tissue and *Nme4* level was successfully knocked down in mouse hepatocytes (Figs. 3B and EV3A,B). AAV-sh*Nme4* mice significantly decreased the liver weights compared with the control AAV-sh*Scramble* mice (Fig. 3C), with no prominent effect in mice whole body weights (Fig. 3D). The serum ALT level, which indicates the level of liver injury, was also significantly decreased in AAV-sh*Nme4* mice (Fig. 3E). AAV-sh*Nme4* infection greatly alleviated hepatic steatosis in HFD-fed mice, as evidenced by the results of H&E staining and Oil Red O staining (Fig. 3F–H). Together, these data indicated Nme4 is essential for lipid accumulation in HFD mice liver and promotes NAFLD progression.

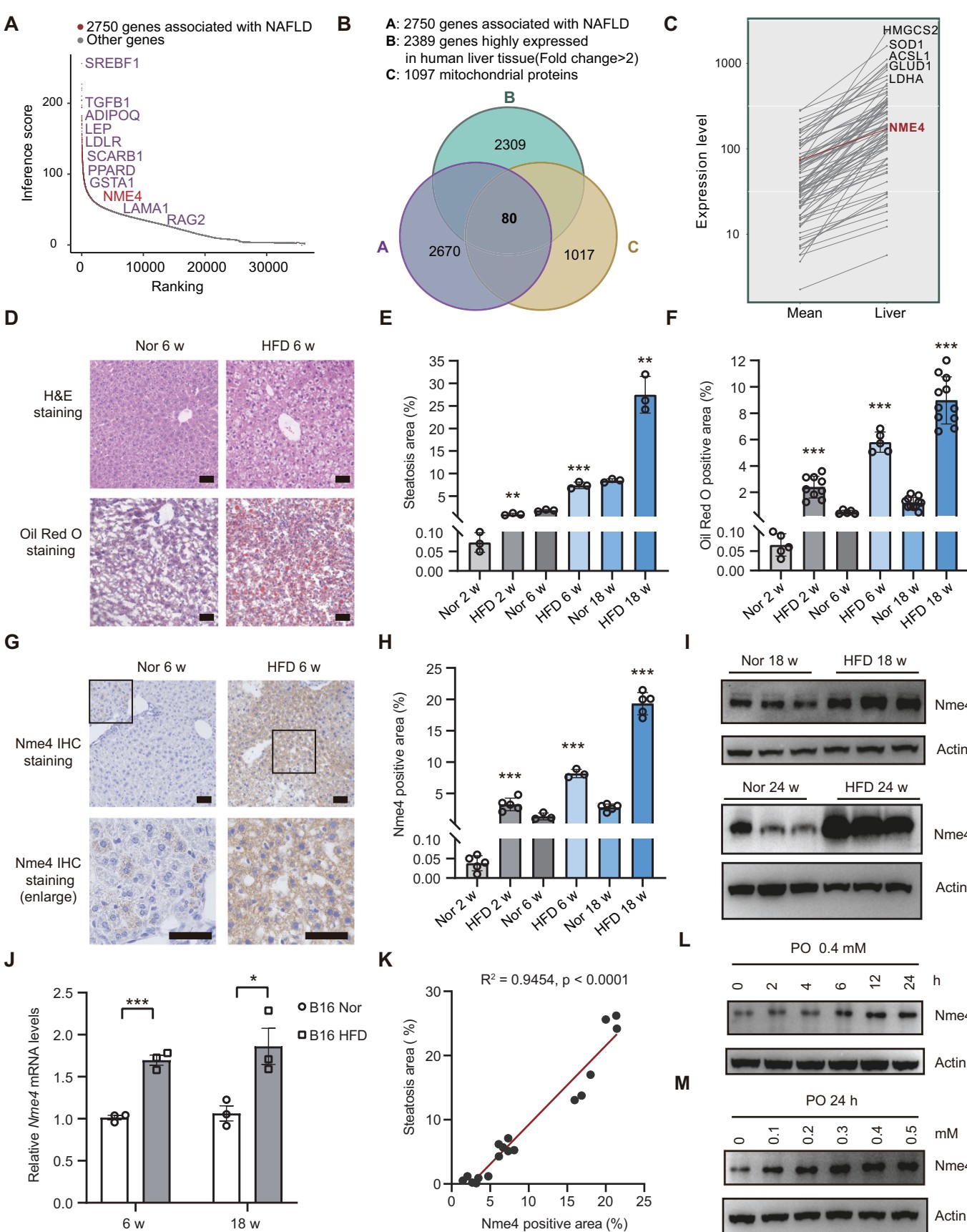

**Figure 1.   NME4 expression is upregulated in fatty liver and correlates with NAFLD progression.**

(A) Database analysis of 2750 genes associated with NAFLD and their rankings. (B) A Venn diagram showing the overlap between the NAFLD genes and mitochondrial proteins that are highly expressed in liver tissue. Eighty proteins, including NME4, were identified. (C) Line chart showing the gene expression in liver compare with mean of all tissue identified in (B). (D) H&E staining and Oil Red O staining of liver sections of the mice fed a HFD for the indicated times. Scale bars, 50 μm. (E) The steatosis area of H&E staining (D) was quantified by Image-Pro Plus (IPP). Biological replicates, n = 3. (F) Oil Red O-positive area of (D) was quantified by Image-Pro Plus (IPP). Biological replicates, in 2 w n > 5; in 6 w n = 5; in 18 w, n = 11. (G) Immunohistochemical staining of liver sections from the mice fed a HFD for the indicated times. Scale bars, 50 μm. (H) Nme4-positive area was quantified by Image-Pro Plus (IPP). Biological replicates, in 2 w n = 5; in 6 w n = 3; in 18 w, n = 5. (I) Western blots probed with antibodies against Nme4 and Actin in the livers of mice fed a normal diet or HFD for 18 weeks and 24 weeks. (J) Relative mRNA levels of Nme4 in the livers of the mice fed a normal diet or HFD for 6 weeks and 18 weeks were measured by RT–qPCR. The mRNA level was detected by qPCR and normalized to β-actin. Biological replicates, n = 3. (K) The correlation between the Nme4-positive area and the steatosis area was analyzed using Pearson's correlation, P values was detected by two-tailed Student's t test (paired). (L,M) Primary hepatocytes were treated with different doses of PO (PA: OA = 0.4 mM : 0.8 mM) for different times. Blots probed with antibodies against Nme4 and Actin are shown. Data information: (E,F,H,J) data are presented as mean ± SEM. *P values ≤ 0.05, **P values ≤ 0.01, ***P values ≤ 0.001 (Student's t test, unpaired). HFD high-fat diet, w week, h hour, PA palmitic acid, OA oleic acid. Source data are available online for this figure.

To explore the mechanism underlying the effect of NME4 on lipid metabolism in hepatocytes, we used lentivirus packaged with Flag-tagged NME4 to overexpress NME4 in hepatocytes which were isolated by two-step perfusion. Oil Red O staining revealed that PO-induced hepatocyte lipid accumulation was markedly exacerbated in NME4-overexpression group (Fig. 3I,J). We also evaluated NME4 expression in human liver cancer cell lines. NME4 expression was relatively high in HepG2 and Bel-7402 cells and low in SMMC-7721 and SK-Hep1 cells (Fig. 3K). Knocking out NME4 in Bel-7402 cells, which express high levels of NME4 (Fig. 3L,M), alleviated PO-induced hepatocyte lipid accumulation (Fig. 3N–Q). The same phenomenon was observed in the mouse Hepa1-6 cell line when Nme4 was depleted (Fig. EV3C–F). In parallel, we overexpressed NME4 in SK-Hep1 cells, which express low levels of NME4 (Fig. EV3G,H). NME4 overexpression markedly promoted PO-induced hepatocyte lipid accumulation (Fig. EV3I–L). Together, these results indicate that NME4 is critical for lipid accumulation in liver cells.

## NME4 promotes lipid accumulation in liver cells by increasing triglyceride levels

To further understand how NME4 impacts hepatic lipid levels, we carried out the untargeted lipidomic analysis in hepatocytes from AAV-shNme4 and AAV-shScramble-infected mice. Nme4 depletion markedly altered the lipidomic landscapes in mouse hepatocytes (Fig. 4A; Dataset EV2). Twenty-six lipids were significantly downregulated in NME4-depleted cells, and 17 lipids were significantly upregulated in AAV-shNme4 infected mice (Fig. 4B). Most of these downregulated lipids were triglycerides (TGs) (Fig. 4C). Targeted lipidomic analysis confirmed that TGs were significantly downregulated in hepatocytes from AAV-shNme4 infected mice (Fig. 4D). In vitro biochemical assays further confirmed that the TGs were significantly decreased in AAV-shNme4 mice (Fig. 4E). We further confirmed these observations in human liver cancer cell lines. Changes in NME4 expression markedly altered the lipidomic landscapes in hepatocytes (Fig. 4F,G; Dataset EV3 and 4). Twenty lipids were significantly downregulated in NME4-depleted cells, and 21 lipids were significantly upregulated in NME4-overexpressing cells (Fig. 4H,I). Most of these lipids were also TGs, while the remainder were diacylglycerol and cholesteryl esters (Appendix Fig. S3A,B). Indeed, knocking out NME4 decreased the cellular TG levels, while its overexpression increased the cellular TG levels in hepatic cells (Fig. 4J,K). In vitro biochemical assays further confirmed that both the TG and

cholesterol levels were significantly decreased in NME4-KO cells treated with PO (Appendix Fig. S3C,D). These data suggest that NME4 promotes lipid accumulation by increasing TG levels in hepatocytes.

## The NME4 protein interaction network revealed its correlation with lipid metabolism and metabolic diseases

To elucidate the mechanism by which NME4 influences TG levels, we conducted tandem affinity purification mass spectrometry analysis (TAP-MS) and proximity-dependent biotinylating with mass spectrometry analysis (TurboID-MS) to identify the proteins with which it interacts (Fig. 5A). TAP-MS has been used to identify proteins that participate in stable interactions (Christianson et al, 2012; Li et al, 2017; Sowa et al, 2009), while proximity labeling, which utilizes biotin ligase to biotinylate nearby proteins, has been used to identify proteins that participate in proximal interaction (Roux et al, 2012). TurboID is more active than the original BioID, and a shorter labeling time is required (Branon et al, 2018). For TAP-MS, we tagged the *C*-terminus of NME4 with SFB triple-tag (S-tag, Flag-tag, and SBP-tag). For TurboID-MS, we tagged the *C*-terminus of NME4 with a TurboID-Flag-tag (Fig. EV4A). To mitigate the influence of proximity labeling enzymes, we additionally created a mutant variant of NME4 (mNME4) that lacks the mitochondrial localized sequence. Both WT and mNME4 also be biotinylated and recognized by specific streptavidin antibody (Fig. EV4B). As expected, NME4 mainly localized to mitochondria (Fig. EV4C,D). The majority of biotinylated protein, which was labeled with TurboID by NME4, was found to localize to the mitochondria. In contrast, mNME4 does not display this behavior, indicating its limited ability to label proteins in close proximity. This suggests that NME4 exhibits good performance in labeling proteins in the vicinity of the mitochondria (Fig. EV4E).

TAP-MS of NME4 identified 3,874 peptides in 707 proteins (Table EV1). We used 74 unrelated TAP-MS experiments that were conducted under identical experimental conditions as controls and assigned a quality-associated probabilistic score to each binary interaction using the MUSE algorithm (Li et al, 2017; Li et al, 2016). Using a MUSE score greater than 0.80 as the cutoff, we identified 89 high-confidence interacting proteins (HCIPs) (Table EV2). The identified HCIPs included prey proteins with extensive functional categories (Fig. EV4F). Most of the HCIPs are enzymes, kinases, and peptidases, suggesting their potential role in regulating enzymatic activities in metabolic processes (Fig. EV4G). Indeed, the HCIPs were highly enriched in metabolic diseases (Fig. 5B) and

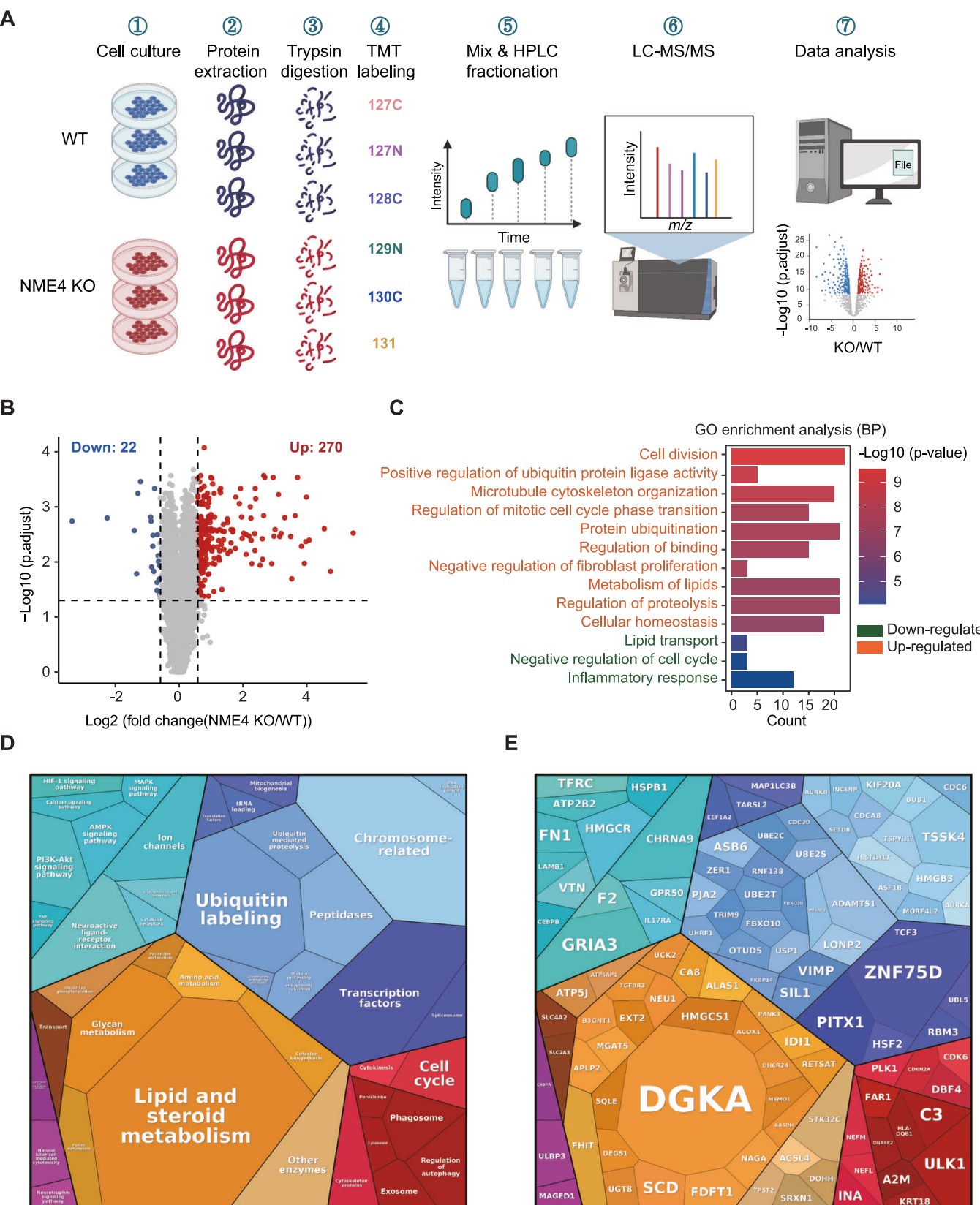

**Figure 2.  Tandem mass tag-based quantitative proteomics revealed that NME4 functions in lipid metabolism.**

(A) Schematic of the tandem mass tag (TMT) 6plex-based quantitative proteomics of WT and NME4-KO cells. (B) Volcano plot showing the levels of proteins in WT and NME4-KO cells. *P* values was estimated by Student's *t* test and adjusted by Benjamini and Hochberg FDR (BH). Biological replicates, *n* = 3. Proteins whose levels were significantly changed are highlighted (fold change > 1.5, *P* values ≤ 0.05). (C) Significantly changed proteins in the NME4 knockout group were enriched in biological processes and analyzed in Hitplot. The top ten biological processes enriched by upregulated proteins was shown. *P* values ≤ 0.05. (D) KEGG pathways of upregulated proteins were mapped using proteomap. (E) Proteins were mapped to the corresponding pathways using proteomap. (D, E) The area of the polygon represents the protein abundance, which was normalized by protein size. Similar colors indicate similar or closely related pathways and proteins. Source data are available online for this figure.

liver complications, such as liver steatosis, liver hyperplasia, and hepatocellular carcinoma (Fig. 5C). To further understand the biological relevance of these proteins, we also performed pathway enrichment using the HCIPs (Fig. 5D,E). Four pathway networks were highly enriched (Fig. 5D), and the related genes are shown (Fig. 5E). Acyl-CoA is the first molecule in TG biosynthesis (Liu et al, 2012), and its dysfunction leads to metabolic diseases, such as hypoglycemia and NAFLD. These results indicated that NME4 interacts with genes involved in TG biosynthesis.

The TurboID-MS data revealed 2441 proximity interactions between NME4 and nearby proteins (Table EV3). We scored the interactions with MUSE using 12 TurboID-MS with random baits as controls and identified 147 HCIPs (Table EV4). For mNME4, we investigated the interactions of mNME4 with MUSE and identified a total of 442 HCIPs (Table EV5) out of 5082 proximity interactions (Table EV6). 45 proteins were identified in both HCIP groups and were used to establish the NME4 interaction network (Fig. 5F). GO enrichment analysis of subcellular complexes showed that the HCIPs were mainly located in the mitochondrial matrix, mitochondrial inner membrane, and mitochondrial protein-containing complexes (Fig. 5G). KEGG enrichment analysis revealed that the HCIPs of NME4 were related to NAFLD (Fig. 5H). In order to investigate the function of mNME4, we conducted GO and KEGG enrichment analysis. We found that the HCIPs of mNME4 are localized throughout the entire cell and are involved in the RNA cycle and DNA repair processes (Appendix Fig. 4A,B; EV4H). Notably, the behavior of mNME4 differs significantly from that of NME4, which suggests that the HCIPs of NME4 are functionally relevant.

## NME4 regulates coenzyme A metabolism and lipid accumulation by interacting with the key enzymes in the pathway

ALKBH7, ACSF3, MLYCD, and CRAT have been reported to be direct regulators of NAFLD (Derdak et al, 2013; Solberg et al, 2013; Sun et al, 2020; Wallace et al, 2018; Wang et al, 2014a), while NUDT19, MMUT, and ALAS1 have been thought to be associated with lipid metabolism (Gorigk et al, 2022; Lian et al, 2018; Luciani and Devuyst, 2020; Wang et al, 2014a; Yao et al, 2018). To understand their role in lipid metabolism in detail, we drew a metabolic pathway based on existing knowledge (Fig. 6A). Thus, NME4 may bind to and regulate the key enzymes catalyzing CoA metabolism (KECCAM) to influence the CoA and TG levels. To confirm this hypothesis, we tagged the C-termini of these candidates with SFB-tags, and then, a pull-down experiment was performed after the transient transfection of the HEK293T cells. All of these candidates bind to NME4 in vitro (Figs. 6B and EV5A,B). Immunofluorescence staining revealed that all the candidates co-localized well with NME4 to mitochondria (Fig. 6C). These data

suggested that NME4 can bind with proteins that are involved in fatty acid catabolic processes.

Malonyl-CoA and acetyl-CoA are the key intermediary metabolites in fatty acid synthesis and provide the primary carbon source for the formation of palmitate (Roduit et al, 2004). To explore the effect of NME4 on acetyl-CoA and malonyl-CoA, targeted metabolomics was performed. The acetyl-CoA and malonyl-CoA levels were significantly reduced both in the NME4-KO cells after PO treatment (Fig. EV5C,D) and in AAV-sh*NME4* mice (Fig. 6D,E). To further understand the details in increased malonyl-CoA production, we performed a lipogenic flux assay utilizing a [2-$^{13}$C] malonic acid isotope tracer, in hepatocytes from control and Nme4 knocked down mice. Nme4 depletion led to significant decreased production of malonyl-CoA and acetyl-CoA derived from isotope-labeled malonate in mouse hepatocytes (Fig. 6F).

NME4 possesses the ability to facilitate the transfer of NTP to NDP while maintaining their equilibrium. Is NME4 involved in the generation of acetyl-CoA through ATP influence? To solve this problem, we detected ATP content in mice tissues and cells. Loss of NME4 has no prominent effect on the level of ATP (Figs. 6G and EV5E,F). ChREBP and SREBP1c are crucial transcription factors responsible for the regulation of genes involved in lipogenesis. The downregulation of NME4 leads to a decrease in the expression of ChREBP and SREBP1c in both cellular and liver tissues (Figs. 6H and EV5G). Furthermore, the qRT–PCR results revealed that NME4 KO downregulated the expression of genes related to de novo lipogenesis and TG synthesis, but it did not affect genes related to TG breakdown both in vitro and in vivo (Figs. 6I and EV5H). These data indicated that NME4 regulates acetyl-CoA and malonyl-CoA levels by interacting with the involved enzymes.

NME4 localizes to mitochondria through a mitochondrial-specific targeting signal in its N-terminus (Milon et al, 2000). To test if NME4 performs its function as a key regulator of NAFLD by binding to these key enzymes, we constructed a mito-deletion plasmid of NME4, which can no longer localize to mitochondria (Fig. 7A). The NME4 mutant no longer bind to KECCAM (Fig. 7B). Re-expressing full-length NME4 in NME4-KO cells significantly increased the TG and TC levels, while expressing the NME4 mito-deletion mutant failed to do so (Fig. 7C,D). The area of Oil Red O staining was also increased after NME4 re-expression in NME4-KO cells, but not after NME4 mito-deletion mutant expression (Fig. 7E,F). These data suggested that NME4 upregulates the TG and TC levels by interacting with the key enzymes in fatty acid catabolic processes.

Taken together, we found NME4 binds to key enzymes catalyzing CoA metabolism, increases malonyl-CoA and acetyl-CoA levels, and subsequently promotes de novo lipogenesis and TG generation, ultimately leading to lipid accumulation and accelerating NAFLD progression (Fig. 7G).

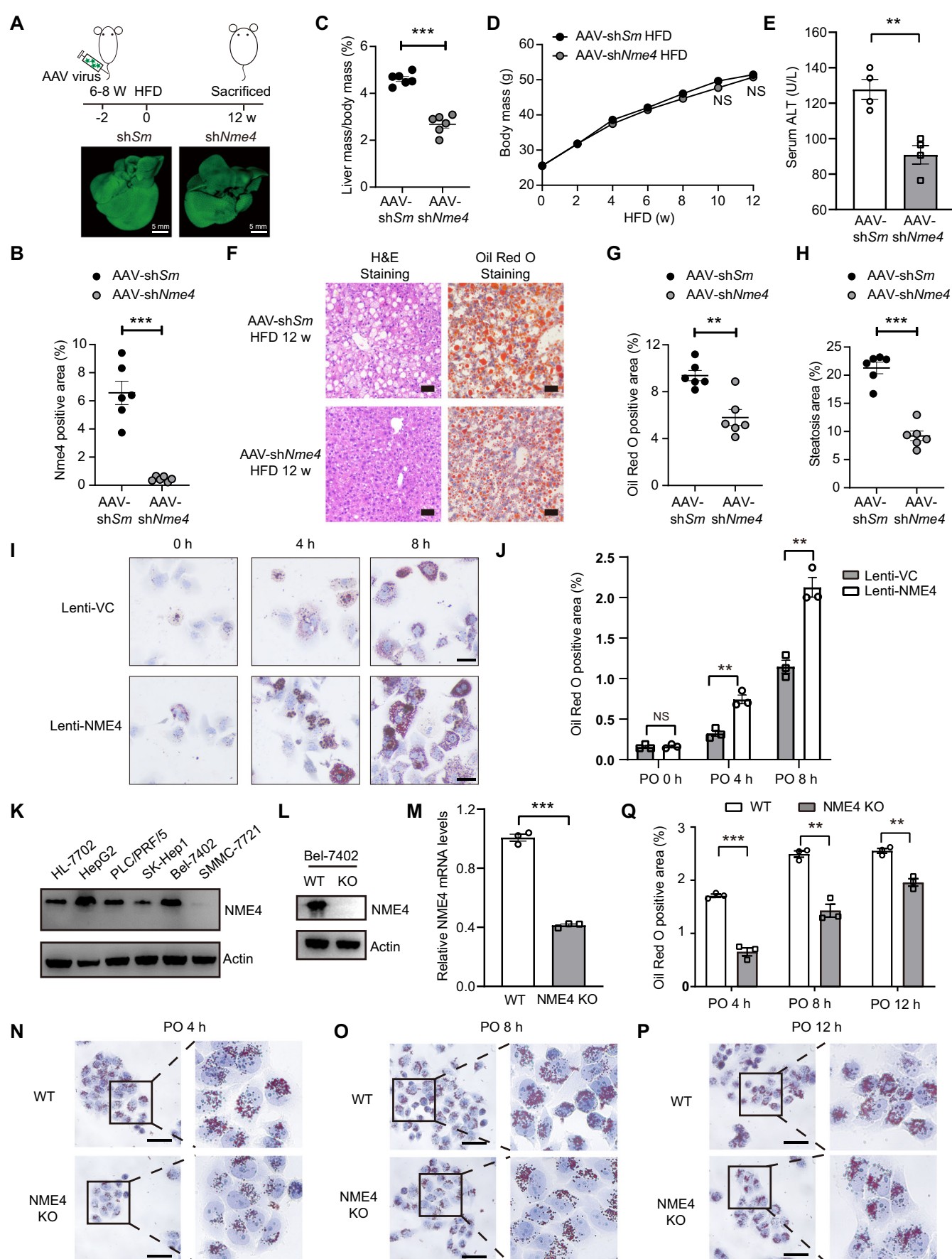

**Figure 3. NME4 promotes lipid accumulation in mouse liver and human liver cell lines.**

(A) Upper is the schematic diagram of Nme4 KD mouse model construction, lower is liver fluorescence imaging with 3 weeks injection of AAV. Scale bars, 5 mm. (B) Nme4 levels in hepatocytes were evaluated in AAV-sh*Nme4* and AAV-sh*Scramble* mice. Biological replicates, $n = 6$. (C) Ratios of mice liver mass to body mass. Biological replicates, $n = 6$. (D) Weight gain curves of mice fed with HFD. Biological replicates, $n = 6$. (E) Serum ALT level of mice liver tissue. Biological replicates, $n = 4$. (F) Representative H&E (left) and Oil Red O (right) staining of liver sections. Scale bars, 50 μm. (G) Steatosis area of H&E staining (F) was quantified by Image-Pro Plus (IPP). Biological replicates, $n = 6$. (H) Oil Red O-positive area of (F) was quantified by Image-Pro Plus (IPP). Biological replicates, $n = 6$. (I) Oil Red O staining of mice hepatocytes which infected with lentivirus expressed with PCDH-VC and PCDH-NME4, then treated with PO for 4 h and 8 h. (J) The Oil Red O-positive areas in (I) were quantified by IPP. Biological replicates, $n = 3$. (K) Western blots probed with antibodies against NME4 and Actin in various liver cell lines. (L) The protein level of NME4 in wild-type and NME4-KO Bel-7402 cells were measured by western blotting probed with antibodies against NME4 and Actin. (M) The mRNA level of NME4 in wild-type and NME4-KO Bel-7402 cells were measured by RT–qPCR. Biological replicates, $n = 3$. (N–P) Oil Red O staining of wild-type and NME4-KO Bel-7402 cells treated with PO for the indicated times. Scale bars, 50 μm. (Q) The Oil Red O-positive areas in N–P were quantified by Image-Pro Plus (IPP). Biological replicates, $n = 3$. Data information: (B–E,G,H,J,M,Q), data are presented as mean ± SEM. \*\**P* values ≤ 0.01, \*\*\**P* values ≤ 0.001, NS -P values > 0.05 (Student's *t* test, unpaired). HFD high-fat diet, Sm Scramble, w week, VC vector, h hour, KO knockout. Source data are available online for this figure.

## Discussion

In this study, we found that NME4 is upregulated in NAFLD and that its expression is positively correlated with liver steatosis. Proteomics and interaction network studies of NME4 reveal that it interacts with several key enzymes involved in CoA metabolism, such as CRAT and NUDT19. Depletion of NME4 reduces the cellular level of acetyl-CoA, blocks de novo lipogenesis, and alleviates lipid accumulation by decreasing TG levels in hepatocytes. These findings highlight the potential function of NME4 in NAFLD progression.

The mechanism by which hepatocytes sense energy stress and activate lipogenesis remains largely elusive. We found that NME4 might be the missing link of this chain. NME4 belongs to the NDPK family, localizes to mitochondria, where NME4 has been proposed to function in the synthesis of ATP or nucleoside diphosphate (NTP) to promote protein and nucleic acid synthesis (Gordon et al, 2006). Our interactome data also supported this role of NME4 in NTP biosynthesis (Fig. EV4F). NME4 also appears to play a very important role in cardiolipin signaling to promote cellular and mitochondrial quality control (MacVicar and Langer, 2016; Tokarska-Schlattner et al, 2008). It is possible that excessive energy stress activates cardiolipin signaling, which in turn activates NME4.

NME4 promotes de novo lipogenesis potentially through the direct binding and activation of key enzymes involved in CoA metabolism, ultimately leading to lipogenesis and steatosis in NAFLD. Among these NME4-binding key enzymes, ACSF3 has been reported to participate in the regulation of fatty acid activation and the synthesis of acetyl-CoA, serving as an important regulatory factor in fatty acid metabolism (Sloan et al, 2011). A previous study has found that the loss of SIRT1 leads to an increase in the level of ACSF3 protein. This increase is believed to be due to the influence of SIRT1 on protein stability, which is regulated by acetylation (Sun et al, 2020). The fluctuation of key enzyme protein levels can lead to disruptions in lipid synthesis and abnormal fatty acid metabolism. Through database analysis, it has been determined that there is a positive correlation between the level of NME4 and CoA metabolism enzymes in liver tissue. This finding suggests that NME4 may play a regulatory role in determining the abundance of these enzymes. Nevertheless, the functional relevance of the interactions between NME4 and the CoA release enzymes has yet to be explored.

NME4 is an intermediate in histidine phosphorylation. NME4 has the capability to transfer phosphate groups to other proteins (Adam et al, 2020; Fuhs and Hunter, 2017b). One hypothesis is that NME4 may phosphorylate these enzymes and subsequently improve protein stability. Further research will be also needed to explore the upstream events of NME4 activation under physiological and pathological conditions.

Despite extensive studies of the mechanisms involved in NAFLD initiation and progression, to date, no specific pharmacological agent has been approved for the treatment of NAFLD. An ideal drug for NAFLD treatment should reduce hepatic steatosis without disrupting the general metabolic processes in mitochondria. We found that depletion of NME4 in hepatocytes blocks de novo lipogenesis and alleviates lipid accumulation without exerting prominent effects on general mitochondrial functions or redox levels. Thus, NME4 is potentially a promising target for the treatment of NAFLD. Our findings may facilitate the development of targeted therapeutic strategies against NAFLD. Such strategies might involve in targeted therapy that inhibits the expression or the protein level of NME4 or combination therapy with metformin, which alters cellular energy uptake and redox state.

Taken together, these results provide a glimpse into the critical role of NME4 in the biosynthesis of TG and lipogenesis, and the progression of NAFLD, shedding light on the development of novel therapeutic strategies for NAFLD and related diseases.

## Methods

### Ethics statement

All animal experiments were conducted according to a protocol approved by the Animal Care and Use Committee of the Westlake University.

### Animals

C57BL/6J mice (Strain #000664), 6–8-weeks-old, were purchased from from Shanghai SLAC Laboratory Animal Company (Shanghai, China) and were bred in the animal facility of Westlake University. All the animal experiments were performed in accordance with a protocol that was approved by the Institutional Animal Care and Use Committee of Westlake University. The mice were maintained on a 12/12 h light/dark cycle at 22–26 °C (humidity, 40–70%) with free access to sterile pellet food and water. All animal studies were approved by the Institutional Animal Care and Use Committee (IACUC) of Westlake University, Hangzhou, China. The local institutional animal ethics board

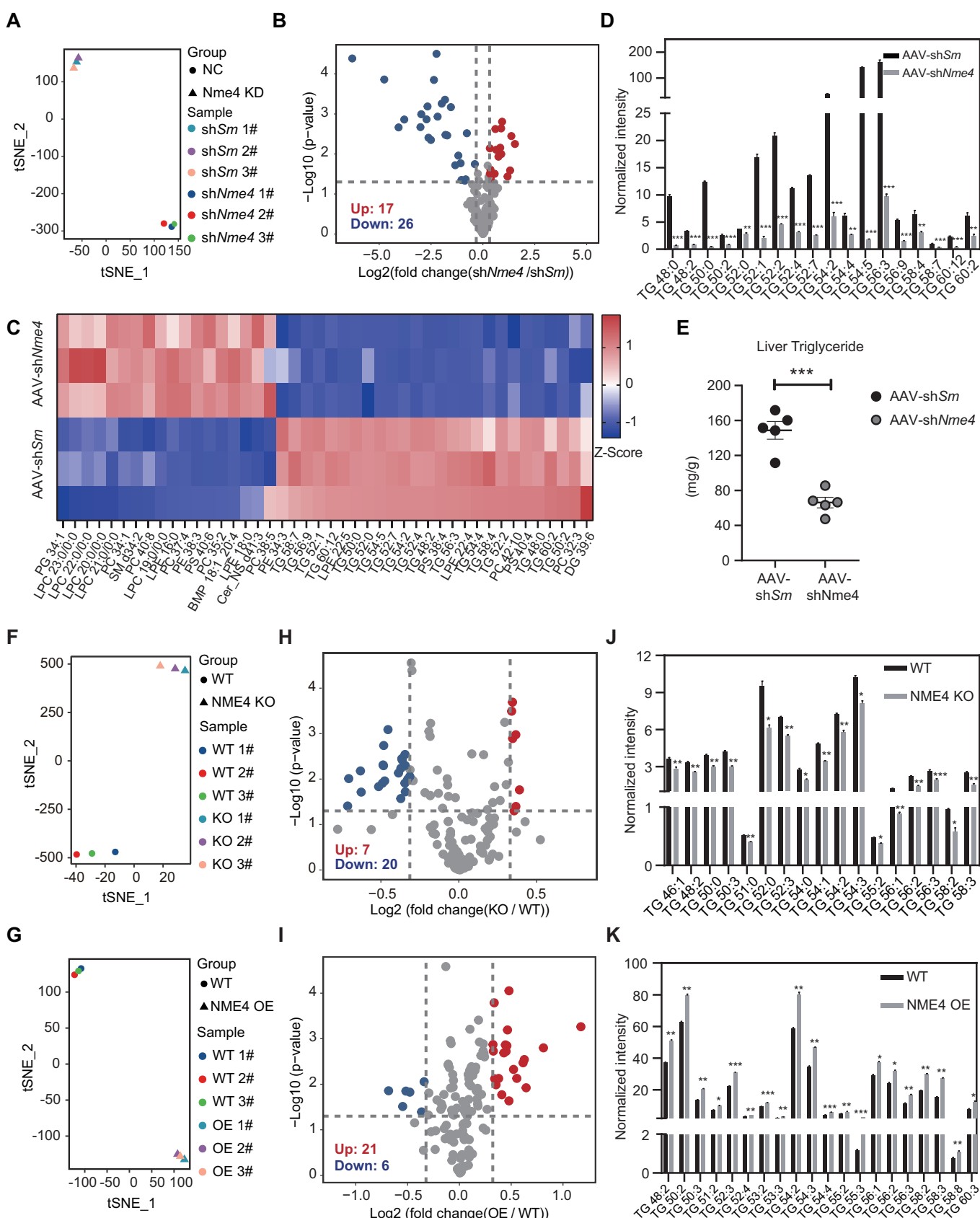

◀ **Figure 4. NME4 promotes lipid accumulation by increasing triglyceride levels in mice liver and human liver cells.**

(A) T-distributed stochastic neighbor embedding analysis of lipidomic data from AAV-sh*Scramble* and AAV-sh*Nme4* mice livers. (B) Volcano plot showing the levels of different lipids in AAV-sh*Scramble* and AAV-sh*Nme4* mice livers. *P* values was done by Student's *t* test. Lipids whose levels were significantly changed were highlighted (fold change >1.25, *P* values ≤ 0.05). Biological replicates, $n = 3$. (C) Heatmap of lipids whose levels were significantly changed in AAV-sh*Scramble* and AAV-sh*Nme4* mice livers. (D) Normalized intensity of different triglyceride (TGs) whose levels were significantly changed in AAV-sh*Scramble* and AAV-sh*Nme4* mice livers. Biological replicates, $n = 3$. (E) AAV-sh*Scramble* and AAV-sh*Nme4* mice livers were separated and triglyceride levels were measured. Biological replicates, $n = 5$. (F,G) T-distributed stochastic neighbor embedding analysis of lipidomic data from wild-type and NME4-KO Bel-7402 cells (F) and wild-type and NME4-OE SK-Hep1 cells (G). (H,I) Volcano plot showing the levels of different lipids in wild-type and NME4-KO Bel-7402 cells (H) and wild-type and NME4-OE SK-Hep1 cells (I). *P* values was done by Student's *t* test. Lipids whose levels were significantly changed were highlighted (fold change >1.25, *P* values ≤ 0.05). Biological replicates, $n = 3$. (J,K) Normalized intensity of different TGs whose levels were significantly changed in wild-type and NME4-KO Bel-7402 cells (J) and wild-type and NME4-OE SK-Hep1 cells (K). Biological replicates, $n = 3$. Data information: (D,E,J,K) data are presented as mean ± SEM. *$P$ values ≤ 0.05, **$P$ values ≤ 0.01, ***$P$ values ≤ 0.001, NS -$P$ values > 0.05 (Student's *t* test, unpaired). Sm Scramble, KO knockout, OE overexpression. Source data are available online for this figure.

approved all mouse experiments (permission numbers: 22-002-LIXU). A mouse model of NAFLD was established by feeding 6- to 8-week-old male C57BL/6J mice a HFD (60 kcal% fat, 20 kcal% protein, 20 kcal% carbohydrate; PD6001; Sysebio, China) for the indicated time periods.

For AAV2/8 transduction in mice hepatocytes, AAV-sh*Scramble*/AAV-sh*Nme4* expressing GFP ($2 \times 10^{12}$ genome copies/mouse; Obio Technology, China) was delivered by tail vein injection for 2 weeks, then followed by 12 weeks of HFD feeding. The mice were then sacrificed for analysis. The mouse *Nme4* shRNA sequence that was used as follows:

5′- CCTCTGTCAACAAGAAGTCAA -3′.

## Cell culture

HEK293T cells (ATCC: CRL-3216), HepG2 cells (ATCC: HB-8065), PLC/PRF/5 cells (ATCC: CRL-8024), and SK-Hep1 cells (ATCC: HTB-52) were purchased from American Type Culture Collection (ATCC, USA). HL-7702, Bel-7402, and SMMC-7721 cells were generous gifts from Dr. Qinfeng Yan, Zhejiang University. Hepa1-6 cells were purchased from Procell Life Science & Technology, China. All the cell lines were maintained in Dulbecco's modified Eagle's medium (DMEM, BasalMedia, China) supplemented with 10% fetal bovine serum (FBS, EXCEL, China) and 1% penicillin and streptomycin (Thermo Fisher Scientific, USA) at 37 °C in 5% $CO_2$ (v/v). The cell lines were routinely tested for mycoplasma contamination and were tested negative.

PO (0.4 mM palmitic acid and 0.8 mM oleic acid) treatment in indicating time was performed according to previous studies (Ge et al, 2022; Li et al, 2021; Wang et al, 2022).

## Plasmid construction, transfection, and lentivirus packaging

Plasmids encoding the indicated genes were amplified from the cDNA of HEK293T cells using RT–PCR. For TAP, all the constructs were subcloned into a pDONOR201 vector using Gateway Technology (Thermo Fisher Scientific, USA) and used as entry clones. The entry clones were subsequently recombined into a gateway destination vector for the expression of S-protein-, 2×Flag-, and streptavidin-binding peptide (SBP)-tandem tag (SFB)-tagged fusion proteins. Gateway-compatible destination vectors with the indicated SFB tag and Myc tag were used to express the various fusion proteins. For TurboID, all the constructs were cloned into the PCDH-MCS-T2A-Puro-MSCV vector using the

ClonExpress® II One Step Cloning Kit (Vazyme, China). Plasmid transfection was performed using polyethylenimine (PEI, Ott Scientific, USA) as the transfection reagent.

To generate NME4 knockout cells, four distinct single-guide RNAs were designed using the Benchling website (https://benchling.com). The sgRNAs were cloned into a lentiCRISPRv2 vector (Addgene plasmid # 98290) that contains a gRNA scaffold and Cas9, followed by transfection into HEK293T cells with pMD2.G and pSPAX2 to generate lentivirus. Forty-eight hours after transfection, the supernatant was collected and used to infect HEK293T, Bel-7402, and Hepa1-6 cells. Stably transfected cells were selected by incubation with medium supplemented with 2–5 µg/mL puromycin for 2 days. Overexpression or knockout efficiencies were confirmed using western blotting and qRT-PCR analysis. The human NME4-specific guide sequences that were used are as follows: 5′-CAGGCCCAGAGCTCATGTAG-3′ or 5′-GGTCCTGGTAGTGCTCGGCA-3′.

## Western blotting and immunoprecipitation

Whole-cell lysates were prepared by lysing cells with NETN buffer (20 mM Tris-HCl, pH 8.0, 100 mM NaCl, 1 mM EDTA, 0.5% Nonidet P-40) on ice for 30 min and then boiling the lysates in 2× Laemmli buffer in a 100 °C metal bath for 15 min. The lysates were subjected to SDS–PAGE followed by immunoblotting with antibodies against various proteins as indicated. The following antibodies were used: anti-Flag (M2) (F1804-1MG, 1:5000) (Sigma-Aldrich, USA); anti-Streptavidin-HRP (35104ES60, 1:1000) monoclonal antibodies (Yeasen, China); anti-Myc (M20002, 1:5000, RRID: AB_2861172) monoclonal antibody (Abmart, China); anti-Actin (AC026, 1:10,000, RRID: AB_2768234) polyclonal antibody (Abclonal Technology, China); anti-NME4 (clone OTI1A5) (TA501110S, 1:1000) monoclonal antibody (OriGene, USA); anti-CRAT (A6365, 1:500, RRID: AB_2766967) and anti-MUT (A3969, 1:1000, RRID: AB_2765419) polyclonal antibodies (Abclonal Technology, China); anti-UCP1 (ab234430, 1:1000, RRID: EPR23004-34) monoclonal antibody (Abcam, UK).

For immunoprecipitation assays, $1 \times 10^7$ cells were lysed with NETN buffer on ice for 30 min. The lysates were then incubated with 20 µl of conjugated beads (for SFB-tagged pull-down) for 2 h at 4 °C or incubated with antibodies against endogenous proteins for 2 h at 4 °C, followed by the addition of 20 µl of protein A/G agarose and incubation for 2 h at 4 °C. The beads were washed three times with NETN buffer and boiled in 2 × Laemmli buffer. The lysates were subjected to SDS–PAGE followed by WB.

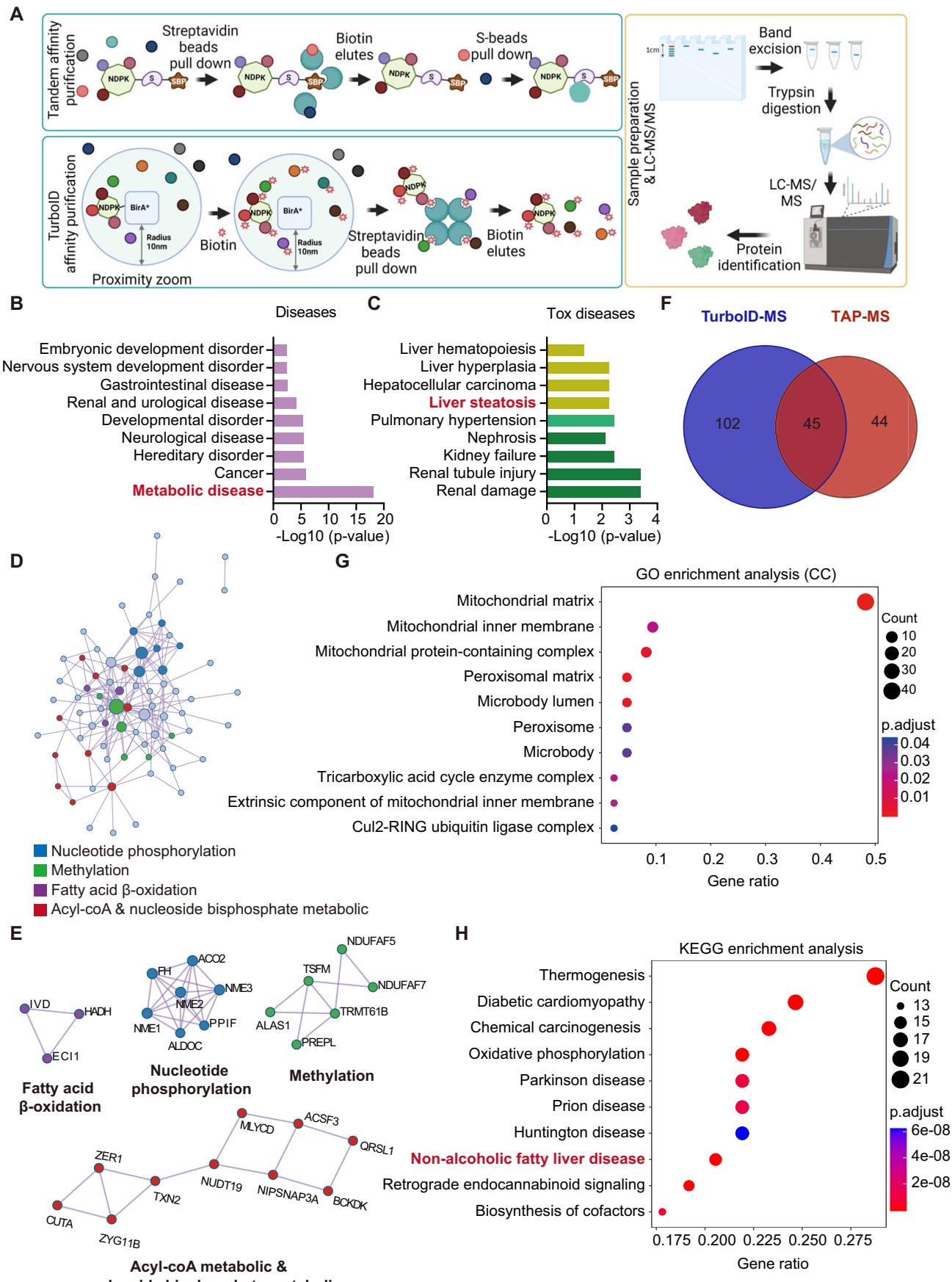

◀ **Figure 5. The NME4 protein interaction network revealed its correlation with lipid metabolism and metabolic diseases.**

(A) Schematic of the integrated proteomics workflow for establishing the NME4 protein interaction network using tandem affinity purification coupled to mass spectrometry (TAP-MS) or TurboID-MS. (B,C) General (B) and specific (C) diseases were enriched using the high-confidence candidate interacting proteins (HCIPs) that were identified by TAP-MS and performed in IPA software (Ingenuity Pathway Analysis). *P* values ≤ 0.05 was presented. (D) Pathway network enrichment of HCIPs that were identified by TAP-MS. (E) The top four pathways enriched in (D) with related genes are shown. (F) A Venn diagram showing the overlap between the NME4 interaction networks established by TAP-MS and TurboID-MS. (G,H) Gene Ontology analysis in Cellular Components (CC) (G) and KEGG analysis in diseases (H) was carried out using TurboID-MS HCIPs and performed by Hitplot. *P*.adjust ≤ 0.05 was significant, *P* values was adjusted by Benjamini and Hochberg FDR (BH).

## Immunofluorescence

For immunofluorescence assays, cells were cultured on coverslips, fixed with 4% paraformaldehyde at 25 °C for 15 min, and then permeabilized with 0.5% Triton X-100 in PBS for 5 min. After blocking with 5% bovine serum albumin, the cells were incubated with the indicated primary antibodies for 1 h at 25 °C. The cells were then washed with PBS twice and incubated with goat anti-rabbit or goat anti-mouse fluorescently -labeled IgG (1:1000, Abcam, UK) for 1 h. The cells were counterstained with 100 ng/mL 4′,6-diamidino-2-phenylindole (DAPI) staining solution (Sigma-Aldrich, USA) for 2 min to visualize nuclear DNA. The coverslips were mounted onto glass slides with FluorSave™ Reagent (Millipore, USA) and visualized under an Olympus IX73 Microscope Imaging System (Olympus, Japan).

## Purification of NME4-associated protein complexes (TAP and TurboID)

HEK293T cells stably expressing SFB-tagged NDPK proteins were isolated by culturing in medium supplemented with 2 µg/mL puromycin, and their expression of these proteins was validated by immunostaining and western blotting as described (Bian et al, 2021). For TAP, HEK293T stable cells were lysed using 5 mL NETN buffer supplemented with protease and phosphatase inhibitors at 4 °C for 30 min. The crude lysates were centrifuged at 14,000×*g* at 4 °C for 15 min. The supernatants were incubated with 100 µL streptavidin-conjugated beads (GE Healthcare, USA) at 4 °C for 2 h. The beads were then washed three times with 5 mL NETN buffer, and the bound proteins were eluted twice with 1.5 mL NETN buffer supplemented with 2 mg/mL biotin (Sigma–Aldrich, USA) at 4 °C for 2 h. The eluates were incubated with 30-µL S-protein beads (Millipore, USA) at 4 °C for 2 h. The beads were then washed three times with 1 mL NETN buffer and subjected to SDS–APGE. Each sample was run into the separation gel for 1 cm so that the whole bands could be excised as one sample for in-gel trypsin digestion and LC–MS.

For TurboID proximity labeling and affinity purification, HEK293T stable cells were grown to 70% confluence in 10 cm dishes before treatment with 500 µM biotin in the culture medium for 10 min. The cells were collected, and affinity purification was performed as described above.

## LC and MS

LC and MS were performed as described previously (Bian et al, 2022; Bian et al, 2021). Briefly, the excised gel bands described above were cut into approximately 1-mm³ pieces, which were then subjected to in-gel trypsin digestion (Shevchenko et al, 1996) and dried. The samples were reconstituted in 5 µl of high-performance

liquid chromatography solvent A (2.5% acetonitrile and 0.1% formic acid). A nanoscale reverse-phase high-performance liquid chromatography capillary column was generated by packing 5-µm C18 spherical silica beads into a fused silica capillary (100 µm inner diameter × ~20 cm length) using a flame-drawn tip. After the column was equilibrated, each sample was loaded onto the column using an autosampler. A gradient was formed, and the peptides were eluted with increasing concentrations of solvent B (97.5% acetonitrile and 0.1% formic acid).

As the peptides were eluted, they were subjected to electrospray ionization and then analyzed by an Orbitrap Fusion Lumos Tribrid Mass Spectrometer (Thermo Fisher Scientific, USA). The source was operated at 1.9 kV, with no sheath gas flow and with the ion transfer tube at 350 °C. The data-dependent acquisition mode was used. The survey scan was conducted from *m/z* 350 to 1500, with a resolution of 60,000 at *m/z* 200. The 20 most intense peaks with charge states of 2 and greater were acquired with collision-induced dissociation with a normalized collision energy of 30% and one microscan; the intensity threshold was set to 1000. MS2 spectra were acquired with a resolution of 15,000. The peptides were detected, isolated, and fragmented to produce a tandem mass spectrum of specific fragment ions for each peptide.

## TMT6-plex quantitative proteomics

Cells were lysed in RIPA buffer (50 mm Tris-HCl, pH 7.5, 1 mm EDTA, 150 mm NaCl, 1% N-octylglycoside, 0.1% sodium deoxycholate, complete protease inhibitor mixture) and incubated for 30 min on ice with occasional vortexing. The crude lysates were cleared by centrifugation, and protein concentrations were measured by BCA assay (Thermo Scientific, USA). One hundred micrograms of protein sample were reduced with DTT, alkylated with iodoacetamide, and precipitated by acetone precipitation at −20 °C for 2 h. The precipitated protein samples were collected by centrifugation, washed with ethanol twice, redissolved in 200 µl 0.1 M TEAB buffer supplemented with 2 µg trypsin and incubated at 37 °C for 16 h. The samples were then repeatedly frozen and thawed in liquid nitrogen to inactivate trypsin. TMT6-plex Isobaric Label Reagent (90061, Thermo, USA) was added to the polypeptide samples at 25 °C for 1 h, followed by the addition of hydroxylamine and incubation for 15 min at 25 °C. The labeled samples were mixed together, desalted through C18 cartridges, and dried in a SpeedVac.

RP-HPLC was performed. Briefly, the first dimension RP separation by micro-LC was performed on a U3000 HPLC System (Thermo Fisher Scientific, USA) by using a BEH RP C18 column (5 µm, 300 Å, 250 mm × 4.6 mm i.d., Waters Corporation, USA). Mobile phase A (2% acetonitrile, adjusted pH to 10.0 using NH₃·H₂O) and phase B (98% acetonitrile, adjusted pH to 10.0 using NH₃·H₂O) were used to develop a gradient. The solvent gradient

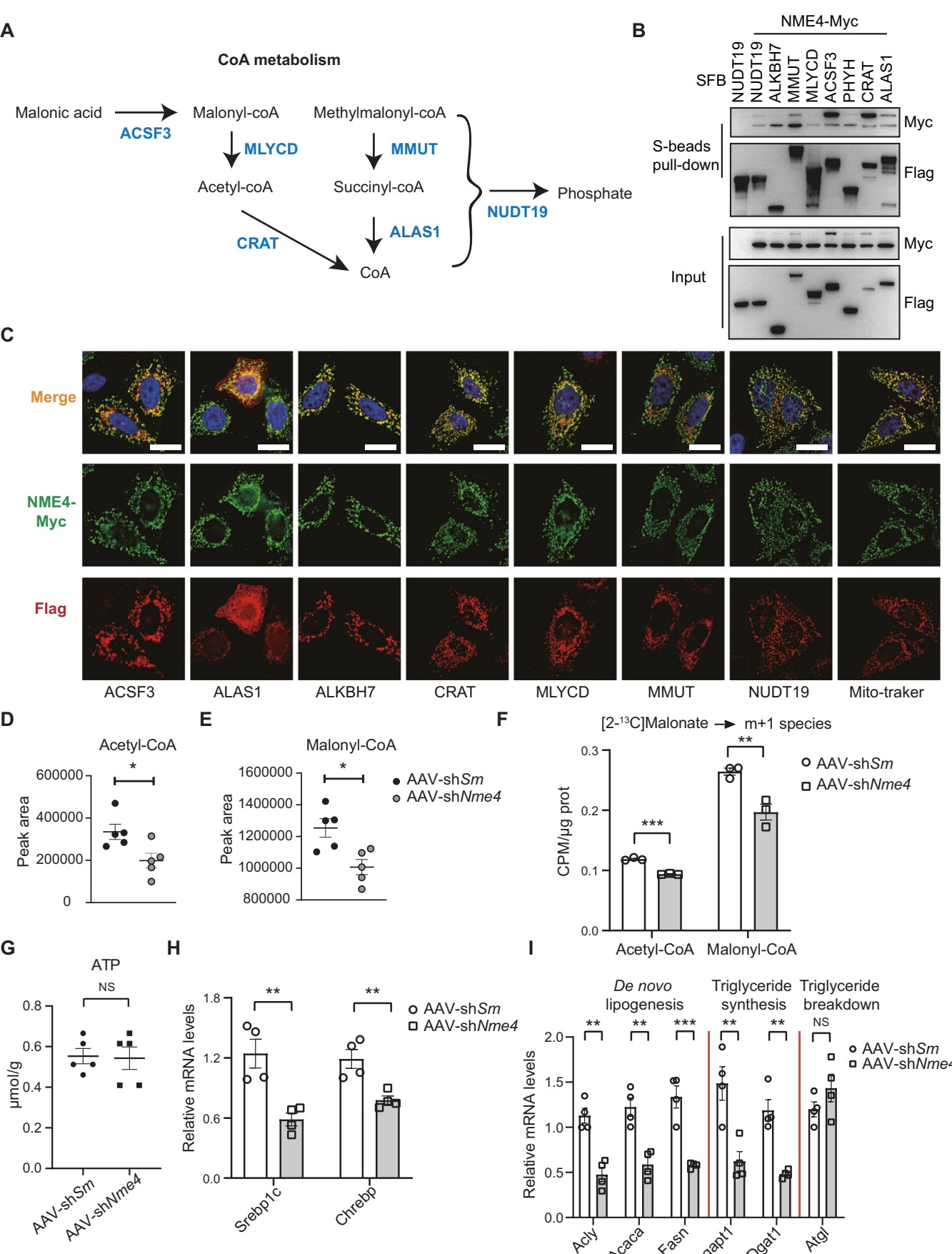

◀ **Figure 6.  NME4 regulates coenzyme A metabolism by interacting with the key enzymes in the pathway.**

(A) Metabolic flow chart shows the CoA-metabolic pathway and the key enzymes catalyzing CoA metabolism (KECCAM). (B,C) HEK293T cells were cotransfected with SFB-tagged KECCAM and Myc-tagged NME4. (B) The cell lysates were incubated with S beads. Five percent lysate was used as the input control. Blots probed with antibodies against the Flag and Myc epitope tags are shown. (C) The cells were subjected to immunofluorescence with an anti-Myc antibody to identify NME4 (green), an anti-Flag antibody to identify SFB-tagged KECCAM or MitoTracker (red), and DAPI (blue) and visualized by microscopy. Scale bars, 20 μm.
(D,E) AAV-sh*Scramble* and AAV-sh*Nme4* mice fed with 12 weeks HFD and livers were separated. The acetyl-CoA (D) and malonyl-CoA (E) levels were determined by targeted metabolite analysis. Biological replicates, $n = 5$. (F) Primary hepatocytes were treated with [2-$^{13}$C] Malonate were extracted. The acetyl-CoA (m + 1) and malonyl-CoA (m + 1) levels were determined by targeted metabolite analysis. Biological replicates, $n = 3$. (G) AAV-sh*Scramble* and AAV-sh*Nme4* mice fed with 12 weeks HFD and livers were separated. The ATP levels was measured by kit. Biological replicates, $n = 5$. (H) Relative mRNA levels of Srebp1c and Chrebp were measured by RT–qPCR. Biological replicates, $n = 4$. (I) Relative mRNA levels of key genes involved in de novo lipogenesis, triglyceride synthesis, and triglyceride breakdown were measured by RT–qPCR. Biological replicates, $n = 4$. Data information: (D–I) data are presented as mean ± SEM. *$P$ values ≤ 0.05, **$P$ values ≤ 0.01, ***$P$ values ≤ 0.001, NS $P$ values > 0.05 (Student's $t$ test, unpaired). Sm Scramble. Source data are available online for this figure.

was set to follows: 5–8% B, 1 min; 8–50% B, 6 min; 50–80% B, 14 min; 80–95% B, 1.5 min; 95% B, 7 min; 95–5% B, 2 min; 5% B 5 min. The tryptic peptides were eluted and separated at a flow rate of 1.0 mL/min and monitored at 214 nm. The column oven was set to 45 °C. Eluent was collected every 90 s. Forty fractions were collected, and the samples were dried under vacuum and reconstituted in 15 μl of 0.1% (v/v) formic acid.

Fractions from the first dimension RPLC were dissolved with loading buffer and then separated by a C18 column (75 μm inner diameter, 360 μm outer-diameter × 15 cm, 2 μm C18). Mobile phase A consisted of 0.1% formic acid in water solution, and mobile phase B consisted of 0.1% formic acid in 80% acetonitrile solution; a series of adjusted linear gradients according to the hydrophobicity of fractions eluted in 1D LC with a flow rate of 300 nL/min was applied. The MS conditions were as follows. For Orbitrap Fusion Lumos, the source was operated at 1.9 kV, with no sheath gas flow and with the ion transfer tube at 350 °C. The mass spectrometer was programmed to acquire data in data-dependent mode. The survey scan was from *m/z* 350 to 1500 with a resolution of 60,000 at *m/z* 200. The 20 most intense peaks with charge state 2 and above were acquired with collision-induced dissociation with a normalized collision energy of 30% and one microscan, and the intensity threshold was set at 1000. The MS2 spectra were acquired with 15,000 resolutions. The peptides were detected, isolated, and fragmented to produce a tandem mass spectrum of specific fragment ions for each peptide.

## MS data analysis and bioinformatics analysis

MS peptide sequences and protein identity were determined by matching the fragmentation patterns in protein databases using the Mascot software program (Matrix Science, USA). Enzyme specificity was set to partially tryptic with two missed cleavages. The modifications of the peptides included carboxyamidomethyl (cysteines, variable) and oxidation (methionine, variable). The mass tolerance was set to 20 ppm for both precursor ions and fragment ions. The database searched was Swiss-Prot (*Homo sapiens*). Spectral matches were filtered to maintain the false-discovery rate at less than 1% at the peptide level using the target-decoy method (Elias and Gygi, 2007), and protein inference was considered following the general rules (Nesvizhskii and Aebersold, 2005) with manual annotation applied when necessary. This same principle was used for protein isoforms when they were present. Generally, the longest isoform was reported.

AP-MS data analysis was performed using the MUSE algorithm as described previously (Li et al, 2016, 2017) to assign quality scores to the identified PPIs. Seventy-four unrelated TAP-MS experiments were

conducted under identical experimental conditions and were used as controls for the MUSE analysis. An MUSE score was assigned to each identified interaction, and any interacting protein with an MUSE score of at least 0.8 and raw spectral counts greater than 1 was considered to be an HCIP. TurboID-MS data were compared with the unrelated TurboID-MS experiments under identical conditions.

The overall and individual groups of proteins were enriched in signaling pathways, functional categories and diseases as indicated. *P* values were estimated using the Knowledge Base included with the Ingenuity Pathway Analysis software program (Ingenuity Systems, USA), Gene Ontology (GO), and KEGG pathways. The GO and KEGG analysis were carried out by using R software (v.4.2.2) package clusterProfiler (v.4.5.0) (Wu et al, 2021) through Hiplot Pro (https://hiplot.com.cn/), a comprehensive web service for biomedical data analysis and visualization.

Only statistically significant correlations ($P \leq 0.05$) are shown. The -log (*P* values) for each function and related proteins is listed. Proteomaps that indicate protein abundance and related signaling pathways were drawn by Bionic visualization (Liebermeister et al, 2014). Network enrichment was performed using Metascape (Zhou et al, 2019), which provides a biologist-oriented resource. The 2750 NAFLD genes were downloaded from the Comparative Toxicogenomics Database (Davis et al, 2021). The collection of mitochondrial proteins and their expression pattern in human tissues were downloaded from previous studies (Pagliarini et al, 2008; Uhlen et al, 2015). The single-cell results were downloaded in Tabula Muris database (Tabula Muris et al, 2018).

## Isolation and culture of primary hepatocytes

Primary hepatocytes were isolated from male C57BL/6J mice at 6–8 weeks of age. The mice were anesthetized and perfused with perfusion buffer and collagenase IV (1 mg/mL, Yeasen, China) through the portal vein at 37 °C. The liver of each mouse was cut, dispersed, and filtered through a 70-μm cell strainer (Thermo Fisher Scientific, USA) and centrifuged at 1000×*g* for 5 min at 4 °C. The cells were resuspended in DMEM supplemented with 10% FBS and incubated at 37 °C in 5% $CO_2$. After incubation for 24 h, the hepatocytes were attached to the culture plate, and the other unattached cells were removed by replacing the culture medium.

## Quantitative real-time PCR (qPCR)

qRT-PCR was performed as described previously (Wang et al, 2021). Briefly, total RNA was extracted using TRIzol reagent (Takara, Japan) according to the manufacturer's instructions. The

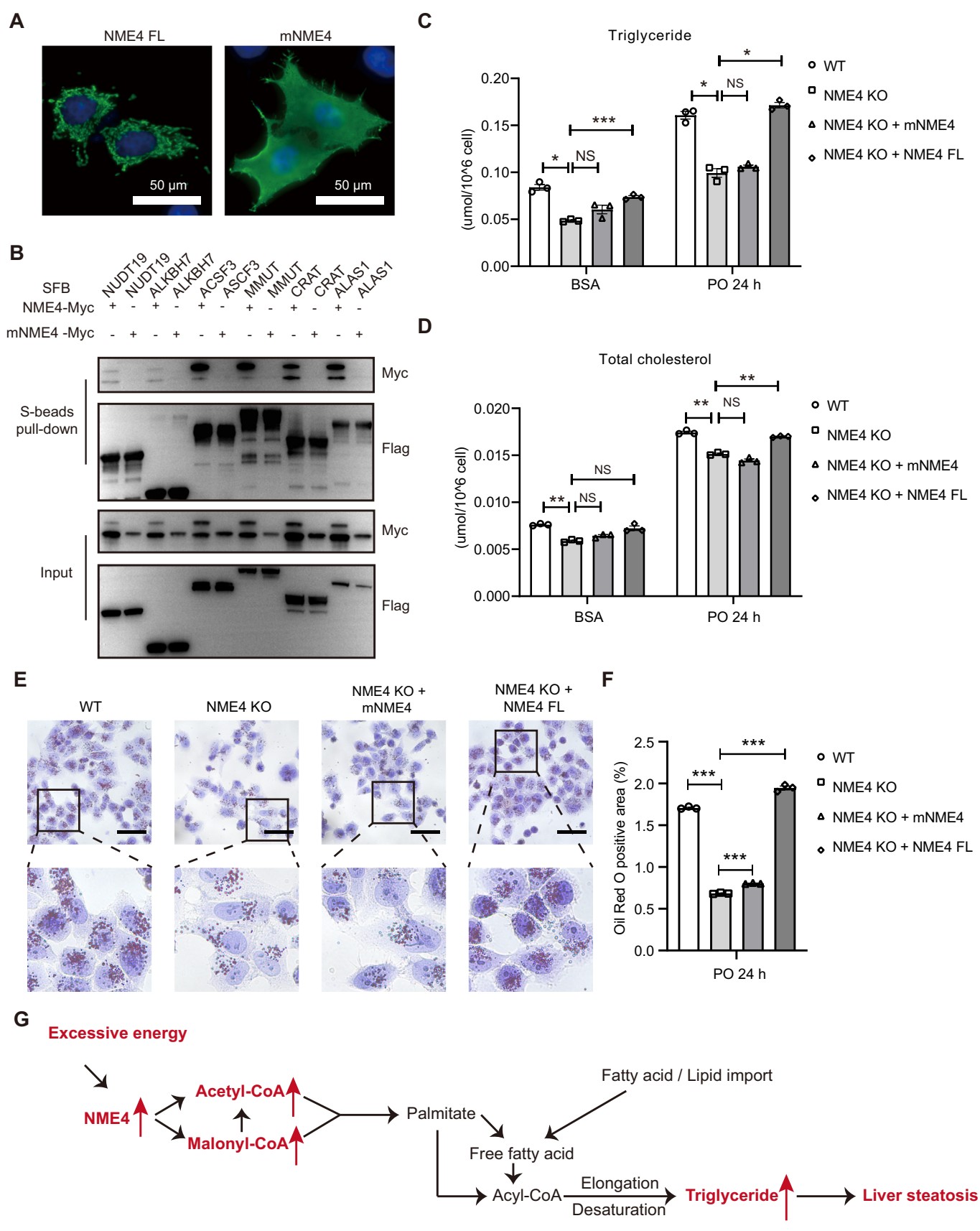

Figure 7. NME4 regulates lipid accumulation by interacting with KECCAM.

(A) HEK293T cells were transfected with Myc-tagged full-length NME4 (FL) or mitochondrial localization sequence-deleted NME4 (mNME4). The cells were subjected to immunofluorescence with an anti-Myc antibody to identify NME4 (green) and DAPI (blue) and visualized by microscopy. Scale bars, 50 μm. (B) HEK293T cells were cotransfected with SFB-tagged KEC-CoA and Myc-tagged NME4 or Myc-tagged mNME4 (mitochondrial localization sequence-deleted in NME4). The cell lysates were incubated with S beads. Five percent lysate was used as the input control. Blots probed with antibodies against the Flag and Myc epitope tags are shown. (C–F) Wild-type, NME4-KO, NME4-KO + NME4 (FL), and NME4-KO+mNME4 (mitochondrial localization sequence-deleted in NME4) Bel-7402 cells were treated with PO for 24 h. (C,D) Triglyceride (C) and total cholesterol (D) levels were measured by related reagent kits. (E) Oil Red O staining was performed in these cells. Scale bars, 50 μm. (F) Oil Red O-positive areas in (E) were quantified by IPP. Biological replicates, $n = 3$. (G) Schematic of the functions of NME4 in lipid metabolism in NAFLD patients. Data information: (C,D,F) data are presented as mean ± SEM. *P values ≤ 0.05, **P values ≤ 0.01, ***P values ≤ 0.001, NS -P values > 0.05 (Student's t test, unpaired). FL full length. Source data are available online for this figure.

concentration of the RNA was determined using a NanoDrop 3000 (Thermo Fisher Scientific, USA). RNA from each sample was reverse-transcribed into cDNA using the HiScript® II 1st Strand cDNA Synthesis Kit (Vazyme, China). The mRNA levels of specific genes were quantified by qPCR using TB Green Premix Ex Taq (Takara, Japan) on a Quant studio 5 qPCR system (Applied Biosystems, USA).

Data were analyzed using the Pfaffl method, based on $\Delta\Delta - Ct$ and normalized to β-actin as the housekeeping gene. The primer that was utilized in this study can be found in Table EV7.

## Oil Red O staining and histological analyses

For Oil Red O staining, cells were cultured in a confocal cell culture dish in the presence of 0.8 mM oleic acid (OA, MCE, USA) and 0.4 mM palmitic acid (PA, MCE, USA) for the indicated time points. The medium was aspirated, and the cells were rinsed twice with PBS, fixed with 4% formaldehyde for 15 min, and dehydrated with 100% 1,2-propanediol for 10 min. Then, prewarmed 0.25% Oil Red O (BBI, China) was added directly to the cells and incubated at 25 °C for 1 h. The residual dye was washed away using 85% 1,2-propanediol followed by ddH$_2$O. The nuclei were stained with hematoxylin (Solarbio Life Sciences, China) at 25 °C for 5 min. The microscopy studies were performed with a microscope equipped with a digital camera. The cells were visualized under an Olympus IX73 Microscope Imaging System (Olympus, Japan).

Mouse liver sections were embedded in paraffin and then stained with hematoxylin (Solarbio Life Sciences, China) to visualize the pattern of lipid accumulation and the inflammatory status. Lipid droplet accumulation was visualized using Oil Red O staining of frozen liver sections that were prepared in Tissue-Tek OCT compound (SAKURA, USA). Paraffin-embedded mouse liver sections were immunostained. After quenching the endogenous peroxidase activity by incubating the sections with 3% H$_2$O$_2$ for 30 min at 37 °C, the sections were processed for antigen retrieval in a microwave and then blocked with 5% BSA for 60 min at room temperature. Primary antibodies were used to detect the expression of the indicated proteins anti-NME4 (ab228005, 1:100) rabbit polyclonal (Abcam, UK). After incubation with horseradish peroxidase (HRP)-conjugated secondary antibodies, the sections were visualized with DAB (Servicebio Technology, Wuhan, China). The slides were stained with hematoxylin, dehydrated, and mounted for bright-field microscopy.

## Metabolite extraction and measurement

Bel-7402 cells were cultured in six-well cell culture plates in the presence of PO for 24 h. The medium was aspirated, and the cells were washed twice with cold PBS. The cells were scraped with 800 μl of cold isopropanol using a cell scraper, and the cell debris and isopropanol were transferred to a 1.5-mL centrifuge tube. The extracts were allowed to incubate at 4 °C for 1 h, followed by 1 min of continuous vortexing. The tubes were centrifuged, and the supernatants were transferred to a new tube and then stored on ice for MS analysis.

For metabolic analysis, Bel-7402 cells were cultivated in six-well cell culture plates in the presence of PO for 24 h. The medium was aspirated as soon as possible, and the cells were washed twice with cold PBS. The cells were scraped with 800 μl cold methanol using a cell scraper and transferred to a new 1.5-mL centrifuge tube. The extracts were incubated at −80 °C for 1 h with occasional vortexing and then centrifuged at 14,000×g at 4 °C for 15 min. The supernatants were transferred to a new tube for MS analysis.

The experiment of Malonyl-CoA and acetyl-CoA generation from Malonate was performed as described previously (Bowman et al, 2017). Briefly, [2-C$^{13}$] Malonate was added to the culture cell and incubated in 37 °C for 4 h. The cells were scraped with 800 μl of cold methanol using a cell scraper and subsequently transferred to a new 1.5-mL centrifuge tube. The extracts were then incubated at −80 °C for 1 h, with intermittent vortexing, before being centrifuged at 14,000×g at 4 °C for a duration of 15 min. The resulting supernatants were carefully aliquoted into a new tube in preparation for mass spectrometry (MS) analysis.

TG and TC contents were analyzed using TG/TC detection kits (Solarbio Life Sciences, China) according to the manufacturer's instructions. TG and TC concentrations were measured by using a microplate reader at 420 nm and 500 nm wavelengths, respectively.

ATP contents were analyzed using ATP detection kits (Solarbio Life Sciences, China) according to the manufacturer's instructions. ATP level was measured by using a microplate reader at 340 nm.

## Statistical analysis and reproducibility

No preprocessing of the data was performed. The sample size for each experiment was determined on the basis of a minimum of $n = 3$ independent devices for each experimental group. All western blotting, immunofluorescence, and RT-qPCR data were obtained from at least three repeated experiments. The data were analyzed using Prism 5.0 tactical significance between two groups was determined by unpaired two-tailed Student's t test. $P \leq 0.05$ was considered statistically significant. *$P \leq 0.05$, **$P \leq 0.01$, ***$P \leq 0.001$, NS -P > 0.05. This study was approved by the Ethics Committee of Westlake University. The operators were not blinded,

the data was collected and analyzed in an objective manner without any bias.

## Data availability

The TMT6 proteomics' raw data of this study have been deposited to the ProteomeXchange Consortium via the PRIDE [https://www.ebi.ac.uk/pride] and assigned the identifier [PXD036628]. The protein interaction raw data of this study have been deposited to the ProteomeXchange Consortium via the PRIDE [https://www.ebi.ac.uk/pride] and assigned the identifier [PXD036624].

## Peer review information

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

## Acknowledgements

We thank Chunqing Song, Liangfeng Wu, Shang Cai, Qingfeng Yan, Aifu Lin, Xiaochun Yu, and Hongtao Yu for their kind help, advice, and critical reading. We thank the Westlake University Supercomputer Center and Biomedical Research Core Facilities for computational resources and related assistance. This work was supported by the Natural Science Foundation of China (32270764, 91954103), the Natural Science Foundation of Zhejiang Province (LR23C070001), the high-risk high-reward program from Westlake Laboratory of Life Sciences and Biomedicine (202209006), and an institutional startup grant from the Westlake Education Foundation.

## Author contributions

**Shaofang Xie**: Validation; Investigation; Visualization; Methodology; Writing—original draft. **Lei Yuan**: Investigation; Methodology. **Yue Sui**: Data curation; Formal analysis; Visualization; Methodology; Writing—original draft. **Shan Feng**: Formal analysis; Investigation; Methodology. **Hengle Li**: Investigation; Methodology. **Xu Li**: Conceptualization; Resources; Supervision; Writing—original draft; Writing—review and editing.

## Disclosure and competing interests statement

The authors declare no competing interests.

# Expanded View Figures

**Figure EV1.  NME4 is upregulated in fatty liver, and its level is correlated with NAFLD progression.**

(A) Heatmap showing the tissue specificity of 80 mitochondrial proteins that were highly expressed in liver tissue. (B,C) H&E staining and Oil Red O staining of liver sections of the mice fed a high-fat diet (HFD) for the indicated times. Scale bars, 50 μm. (D,E) Western blots probed with antibodies against NME4 and Actin in the livers of mice fed a normal diet or a high-fat diet (HFD) for the 2 weeks and 6 weeks. (F) Relative mRNA levels of NME4 in the livers of the mice fed a normal diet or HFD for 2 weeks was measured by RT–qPCR. Biological replicates, $n = 3$. (G,H) Western blots probed with antibodies against NME4, UCP1 and Actin in the subcutaneous white adipose tissue (sWAT), epididymal white adipose tissue (eWAT) and brown adipose tissue (BAT) of mice fed a normal diet or a high-fat diet (HFD) for the 6 weeks. (I–K) Western blots probed with antibodies against NME4, UCP1 and Actin in the sWAT, eWAT, BAT and liver of mice fed a normal diet or a high-fat diet (HFD) for the 24 weeks. Data information: (F) data are presented as mean ± SEM. **$P$ values ≤ 0.01 (Student's $t$ test, unpaired). w, week; HFD high-fat diet, sWAT subcutaneous white adipose tissue, eWAT epididymal white adipose tissue, BAT brown adipose tissue.

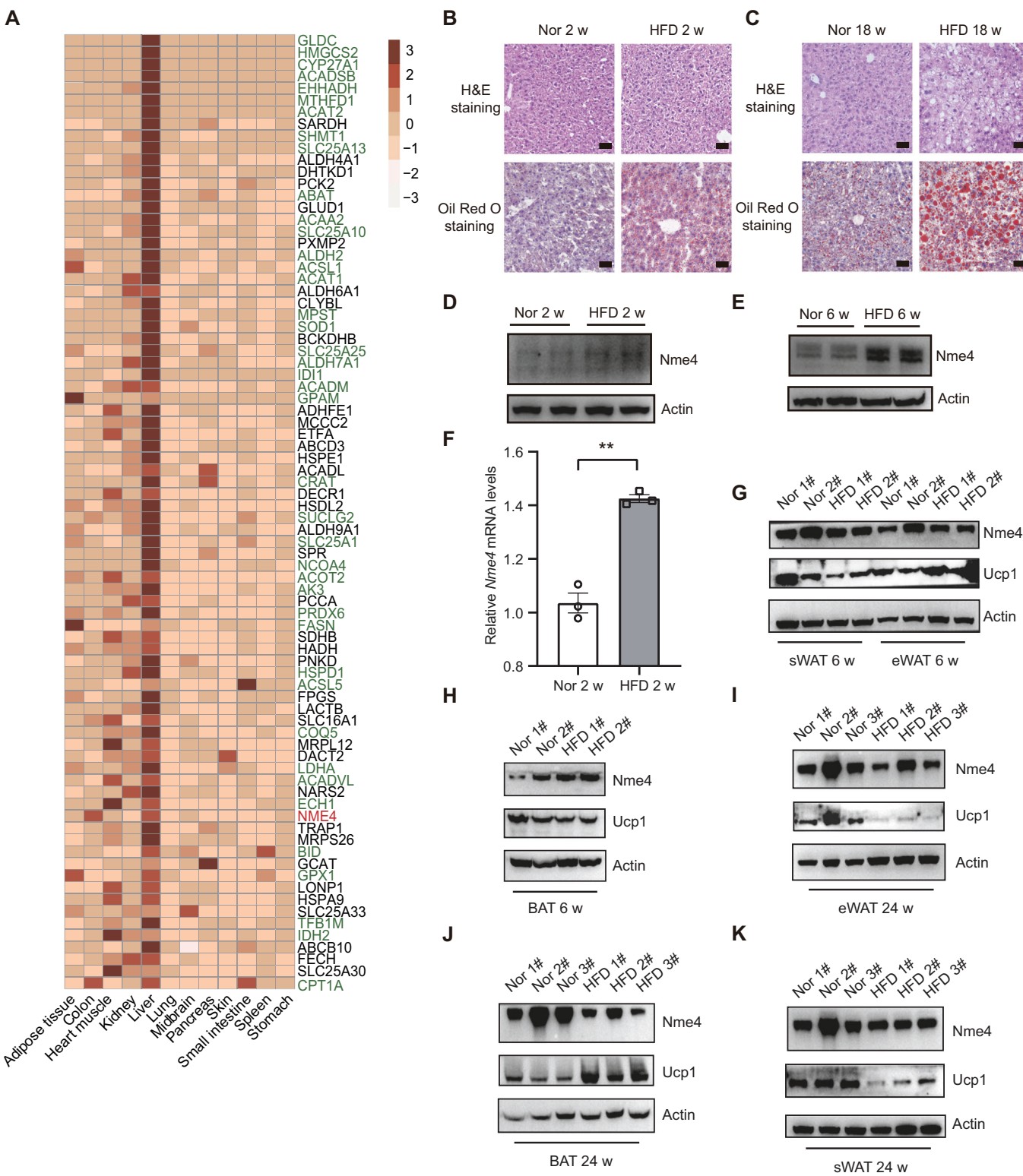

**A**

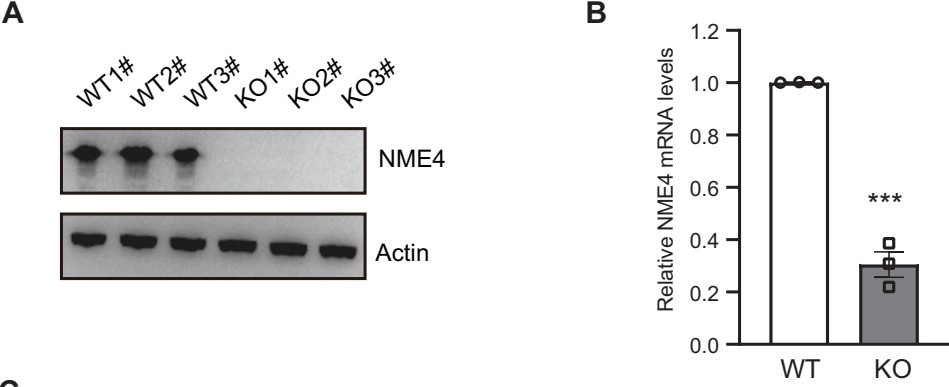

**B**

**C**

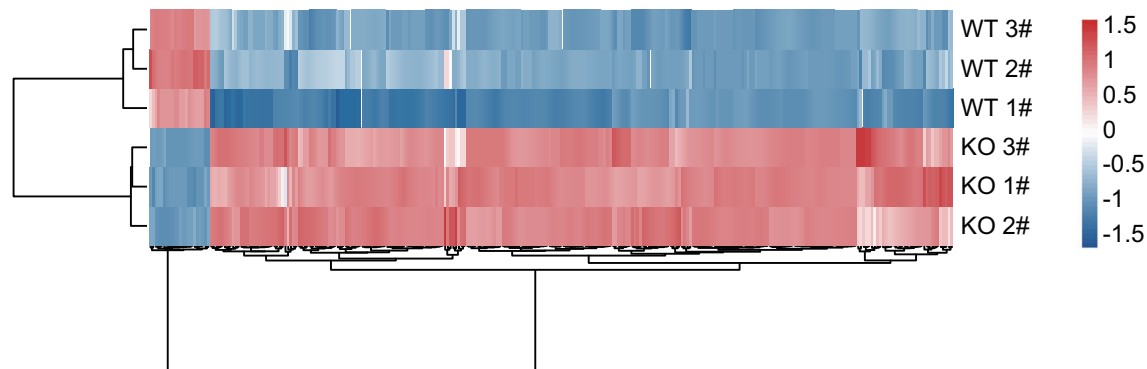

**D**

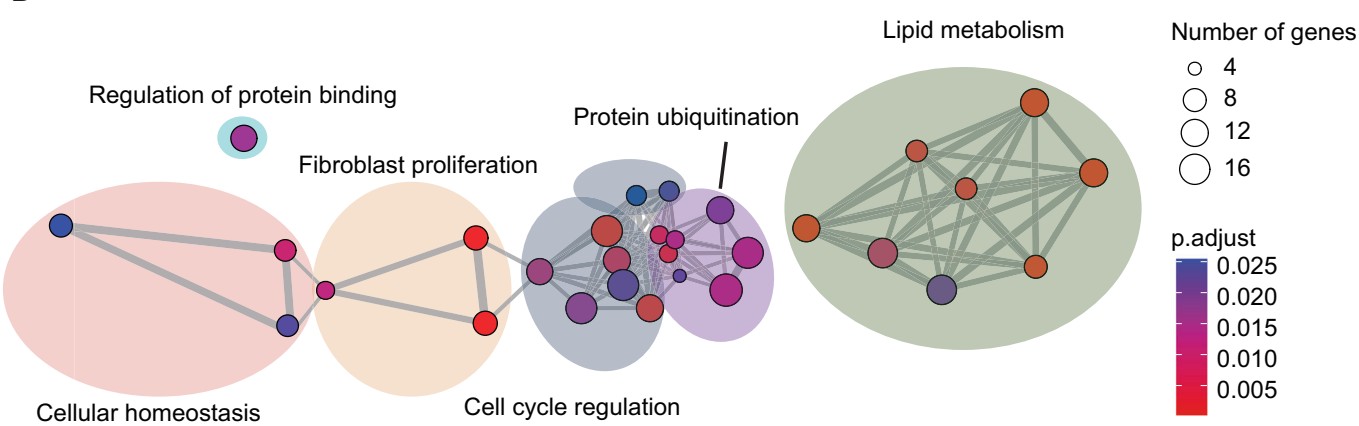

**Figure EV2. Tandem mass tag-based quantitative proteomics reveal a correlation between NME4 and lipid metabolism.**

(A) Representative western blotting (WB) analysis of NME4 expression in HEK293T stable cell lines. (B) Relative mRNA levels of NME4 in HEK293T WT and stable NME4 knockout cell lines. Biological replicates, $n = 3$. (C) Heatmap of proteins with significantly altered expression in TMT6-plex quantitative proteomics. (D) Pathway network enrichment of significantly upregulated proteins identified by TurboID-MS and performed using metascape database. *P*.adjust ≤ 0.05 was cut off which was adjusted by Benjamini and Hochberg FDR (BH). Data information: (B) data are presented as mean ± SEM. ***P* values ≤ 0.001 (Student's *t* test, unpaired). KO knockout.

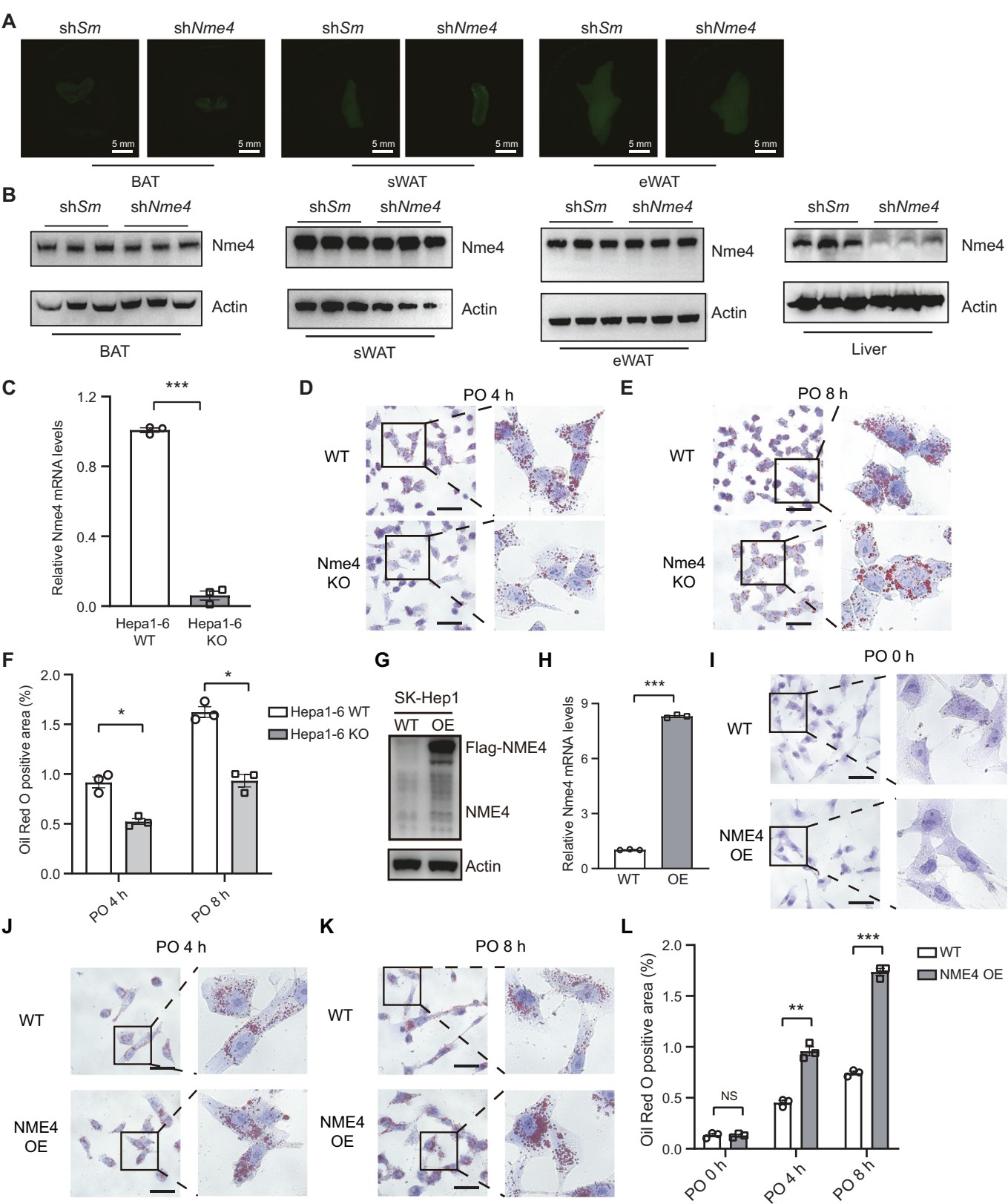

**Figure EV3.   Depleting Nme4 in Hepa1-6 cells reduced lipid accumulation.**

(A) The fluorescence imaging of BAT, sWAT and eWAT from mice with 3 weeks injection of AAV. Scale bars, 5 mm. (B) WB detect the Nme4 and Actin level in BAT, sWAT and eWAT from mice with 12 weeks injection of AAV. (C) The mRNA level of Nme4 in wild-type and Nme4-KO Hepa1-6 cells was measured by RT–qPCR. Biological replicates, $n = 3$. (D,E) Oil Red O staining of wild-type and Nme4-KO Hepa1-6 cells treated with PO for the indicated times. Scale bars, 50 µm. (F) The Oil Red O-positive areas in (B,C) were quantified by IPP. Biological replicates, $n = 3$. (G,H) The protein level (E) and mRNA level (F) of NME4 in wild-type and NME4-OE SK-Hep1 cells were measured by western blotting and RT–qPCR, respectively. Biological replicates, $n = 3$, unpaired Student's $t$ test; ***$P$ values < 0.001. Data are shown as the Mean ± SEM. (I–K) Oil Red O staining of wild-type and NME4-OE SK-Hep1 cells treated with PO for the indicated times. (L) The Oil Red O-positive areas in (G–I) were quantified by IPP. Biological replicates, $n = 3$. Data information: (C,F,H,L) data are presented as mean ± SEM. *$P$ values ≤ 0.05, **$P$ values ≤ 0.01, ***$P$ values ≤ 0.001, NS -$P$ values > 0.05 (Student's $t$ test, unpaired). Sm scramble, sWAT subcutaneous white adipose tissue, eWAT epididymal white adipose tissue, BAT brown adipose tissue, KO knockout, OE overexpression, h hour.

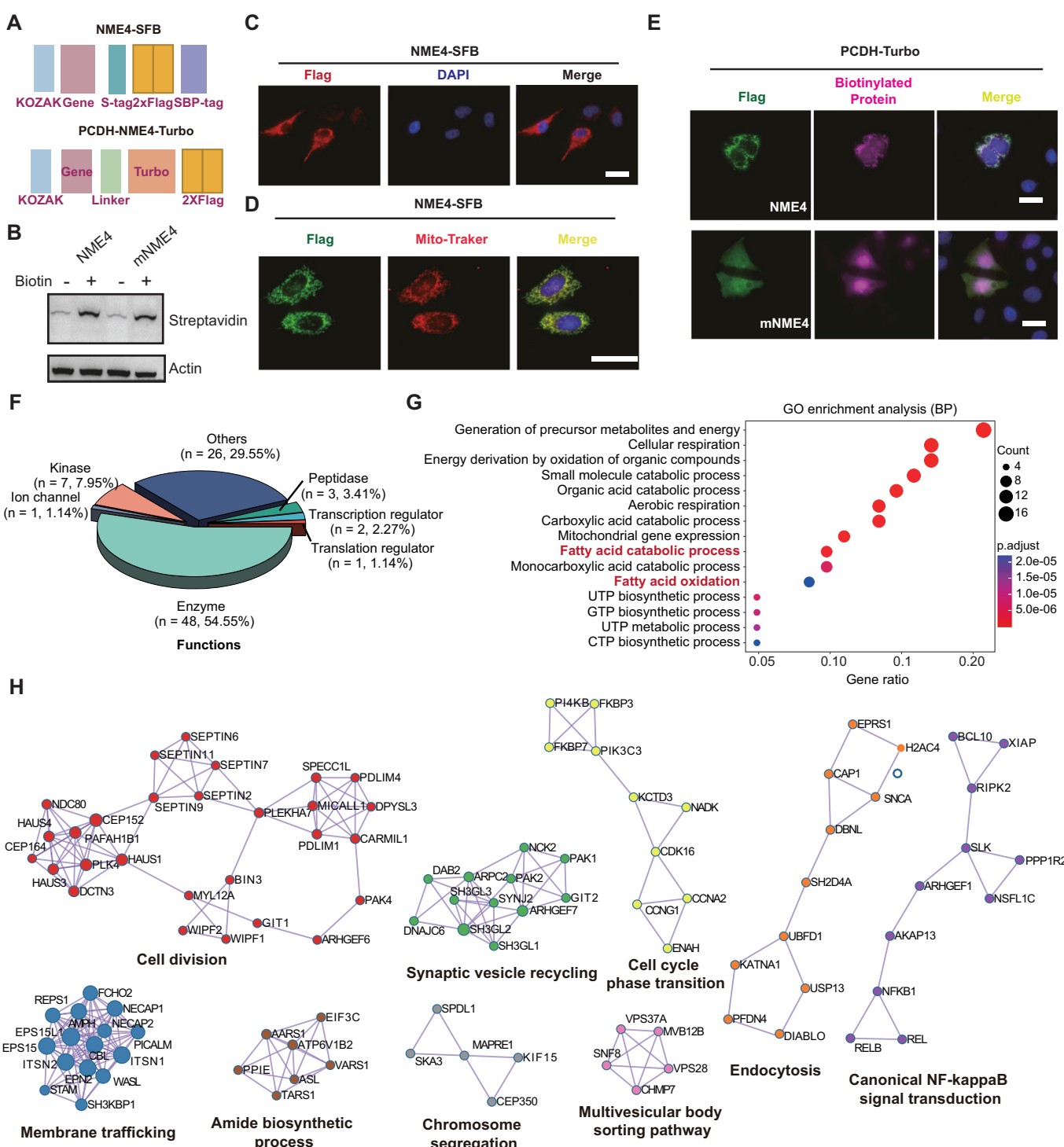

Figure EV4.  **Comprehensive interactome analysis revealed the composition and functions of NME4 interaction networks.**

(A) Plasmid design for NME4 overexpression in TAP-MS and TurboID-MS. (B) HEK293T cells stably overexpressing Turbo-tagged NME4 and mNME4 were subjected to WB after treated with 500 μM biotin incubated in 37 °C for 30 min. Anti-biotinylated protein antibody is specifically recognized biotinylate protein, Actin as the internal control. mNME4, mitochondrial localization sequence-deleted in NME4. (C–E) HEK293T cells stably overexpressing SFB- or Turbo-tagged NME4 were subjected to immunofluorescence with an anti-Flag antibody to identify NME4, MitoTracker is to visualized mitochondrial, anti-biotinylated protein antibody and DAPI and visualized by microscopy. Scale bars, 50 μm. For biotinylated protein labeling, cells were treated with 500 μM biotin for 30 min. (F,G) Results of functional categories (F) and Biological Processes (G) enrichment analysis (GO) of significantly changed proteins in the TAP-MS HCIPs was analyzed in Hitplot. P values ≤ 0.05, here only showed the top 15 of Biological Processes enriched by upregulated proteins. (H) The pathways enriched in mNME4 HCIPs with related genes are shown analyzed in metascape database.

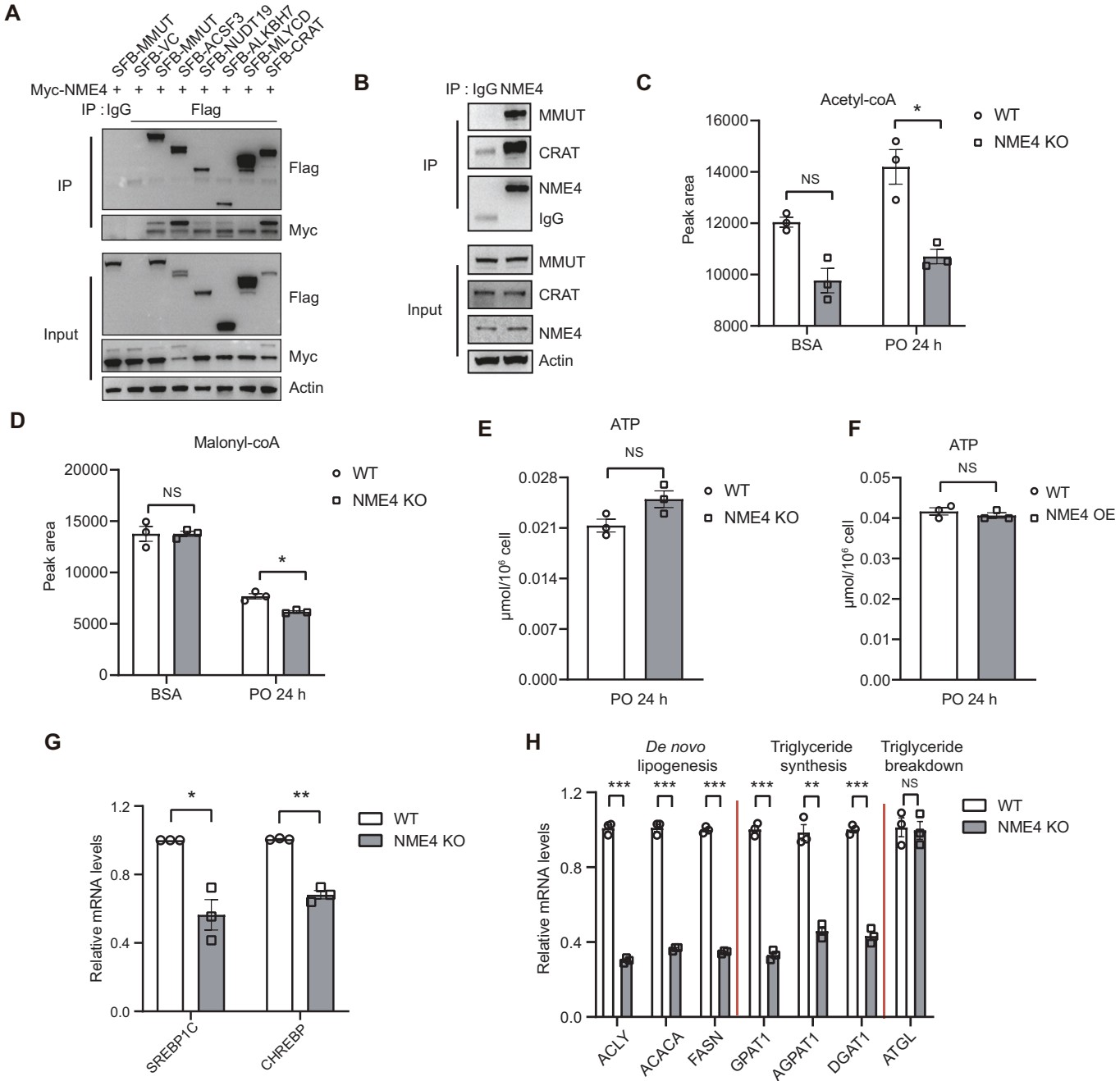

**Figure EV5. NME4 regulates coenzyme A metabolism by interacting with the key enzymes in the pathway.**

(A) HEK293T cells were cotransfected with Myc-tagged NME4 and C-terminal SFB (SFB)-tagged candidate genes, as indicated. The cell lysates were incubated with immunoglobulin G (IgG) control and antibodies against Flag. (B) HEK293T cell lysates were incubated with immunoglobulin G (IgG) control and antibodies against NME4. (C,D) Wild-type and NME4 KO Bel-7402 cells were treated with PO for 24 h. The acetyl-CoA (C) and malonyl-CoA (D) levels were determined by targeted metabolite analysis. Biological replicates, $n = 3$. (E) Wild-type and NME4 KO Bel-7402 cells were treated with PO for 24 h. ATP levels was measured by kit. Biological replicates, $n = 3$. (F) Wild-type and NME4-overexpression SK-Hep1 cells were treated with PO for 24 h. ATP levels was measured by kit. Biological replicates, $n = 3$. (G) Relative mRNA levels of SREBP1C and CHREBP were measured by RT–qPCR. Biological replicates, $n = 3$. (H) Relative mRNA levels of key genes involved in de novo lipogenesis, triglyceride synthesis and triglyceride breakdown were measured by RT–QPCR in wild-type and NME4 KO Bel-7402 cells. Biological replicates, $n = 3$. Data information: (C–H) data are presented as mean ± SEM. *P values ≤ 0.05, **P values ≤ 0.01, ***P values ≤ 0.001, NS -P values > 0.05 (Student's $t$ test, unpaired).

