## [Peer Review File · EMBO Reports]

NME4 Mediates Metabolic Reprogramming and Promotes Nonalcoholic Fatty Liver Disease Progression

shaofang Xie, Lei Yuan, Yue Sui, Shan Feng, Hengle Li, and Xu Li

DOI: 10.15252/embr.202357462

Corresponding author(s): Xu Li (lixu@westlake.edu.cn)

Review Timeline:

Submission Date:	9th May 23
Editorial Decision:	16th Jun 23
Revision Received:	15th Sep 23
Editorial Decision:	12th Oct 23
Revision Received:	8th Nov 23
Accepted:	16th Nov 23

Editor: Deniz Senyilmaz Tiebe

Transaction Report:

Dear Dr. Li,

Thank you for submitting your manuscript to EMBO Reports, which was now seen by three referees, whose reports are copied below.

Referees find the proposed role of NME4 in NAFLD in principle interesting. However, they also raise significant concerns that need to be addressed for publication here.

Regarding the comments of referee #2 (point 6) and referee #3, although we concur with the referees that demonstrating the functional relevance of the interactions between NME4 and the CoA release enzymes would significantly strengthen the manuscript, not being able to do so will not preclude from publication in EMBO Reports, in which case please do make sure to openly discuss this in the manuscript as suggested by referee #3 and mention the alternative mechanisms as per referee #2.

Given the positive recommendations, we would like to invite you to submit a revised manuscript. Please revise your manuscript with the understanding that the referee concerns (as in their reports) must be fully addressed and their suggestions taken on board. Please address all referee concerns in a complete point-by-point response. Acceptance of the manuscript will depend on a positive outcome of a second round of review. It is EMBO reports policy to allow a single round of major experimental revision only and acceptance or rejection of the manuscript will therefore depend on the completeness of your responses included in the next, final version of the manuscript.

We realize that it is difficult to revise to a specific deadline. In the interest of protecting the conceptual advance provided by the work, we recommend a revision within 3 months. Please discuss the revision progress ahead of this time with me if you require more time to complete the revisions, or if you have questions or comments regarding the revision (also by video chat).

1. A data availability section providing access to data deposited in public databases is missing (where applicable).
2. Your manuscript contains statistics and error bars based on $n=2$. Please use scatter plots in these cases.

You can submit the revision either as a Scientific Report or as a Research Article. For Scientific Reports, the revised manuscript can contain up to 5 main figures and 5 Expanded View figures, and it should not exceed 27000 characters. If the revision leads to a manuscript with more than 5 main figures it will be published as a Research Article. In this case the Results and Discussion section should be separate. If a Scientific Report is submitted, these sections have to be combined. This will help to shorten the manuscript text by eliminating some redundancy that is inevitable when discussing the same experiments twice. In either case, all materials and methods should be included in the main manuscript file.

3) We replaced Supplementary Information with Expanded View (EV) Figures and Tables that are collapsible/expandable online. A maximum of 5 EV Figures can be typeset. EV Figures should be cited as 'Figure EV1, Figure EV2" etc... in the text and their respective legends should be included in the main text after the legends of regular figures.

4) a .docx formatted letter INCLUDING the reviewers' reports and your detailed point-by-point responses to their comments. As part of the EMBO publication's Transparent Editorial Process, EMBO reports publishes online a Review Process File (RPF) to

accompany accepted manuscripts. This File will be published in conjunction with your paper and will include the referee reports, your point-by-point response and all pertinent correspondence relating to the manuscript.

<https://www.embopress.org/page/journal/14693178/authorguide#transparentprocess>

5) a complete author checklist, which you can download from our author guidelines

<https://www.embopress.org/page/journal/14693178/authorguide>. Please insert information in the checklist that is also reflected in the manuscript. The completed author checklist will also be part of the RPF.

6) Please note that all corresponding authors are required to supply an ORCID ID for their name upon submission of a revised manuscript (<<https://orcid.org/>>). Please find instructions on how to link your ORCID ID to your account in our manuscript tracking system in our Author guidelines

<<https://www.embopress.org/page/journal/14693178/authorguide#authorshipguidelines>>

Additional information on source data and instruction on how to label the files are available:

<https://www.embopress.org/page/journal/14693178/authorguide#sourcedata>

9) Our journal encourages inclusion of *data citations in the reference list* to directly cite datasets that were re-used and obtained from public databases. Data citations in the article text are distinct from normal bibliographical citations and should directly link to the database records from which the data can be accessed. In the main text, data citations are formatted as follows: "Data ref: Smith et al, 2001" or "Data ref: NCBI Sequence Read Archive PRJNA342805, 2017". In the Reference list, data citations must be labeled with "[DATASET]". A data reference must provide the database name, accession number/identifiers and a resolvable link to the landing page from which the data can be accessed at the end of the reference. Further instructions are available at <http://www.embopress.org/page/journal/14693178/authorguide#referencesformat>

10) Regarding data quantification (see Figure Legends:

<https://www.embopress.org/page/journal/14693178/authorguide#figureformat>)

11) The journal requires a statement specifying whether or not authors have competing interests (defined as all potential or actual interests that could be perceived to influence the presentation or interpretation of an article). In case of competing

interests, this must be specified in your disclosure statement. Further information: <https://www.embopress.org/competing-interests>

12) Please also note our reference format:

I look forward to seeing a revised version of your manuscript when it is ready. Please let me know if you have questions or comments regarding the revision.

Kind regards,

Deniz Senyilmaz Tiebe

Deniz Senyilmaz Tiebe, PhD
Editor
EMBO Reports

Referee #1:

This is an interesting manuscript that reports the effect of NME4 on NAFLD progression.

Xie et al. report that NME4 expression increases in NAFLD and that NME4 inhibition in the liver reduces the susceptibility to High Fat Diet-induced steatosis.

Using a remarkable combination of models and methods, the authors suggest that the mechanisms underlying the effect of NME4 on hepatic lipid metabolism involve binding to a significant number of key enzymes in metabolism.

While, overall, I think this work represents an impressive amount of work, there are several issues the authors may consider to improve the manuscript:

- 1) It seems that NME4 knock-down significantly reduces the expression of lipogenic enzymes in cells. Is it also the case in vivo in the liver of NME4 knock-down? A consistent reduction in the expression of rate-limiting enzymes in fatty acid and TG synthesis could prevent NAFLD. It would also be important to consider the possible effect of NME4 on key lipogenic factors such as ChREBP and SREBP1c.
- 2) I am not sure that NME4 binding to several enzymes involved in metabolic homeostasis is sufficient to modify lipogenesis. This part should be strengthened. One first possible experiment in cell culture would be to measure lipogenic flux in response to change in NME4 modification. In vivo, it would also be interesting to assess the effect of NME4 on the response to a lipogenic challenge (fasting/refeeding high carb) in mice.
- 3) Some of the experiments that have been performed with fatty acids (PO) in cell culture. If I understood the work correctly, the concentration of fatty acids used seems very high (1.2 mM).

Referee #2:

The manuscript of Xie et al characterizes the role of NME in the development of fatty liver disease. They conclude that a maladaptive increase in NME4 expression promotes fatty liver, by NME4 interaction with multiple enzymes that catalyze reactions that release CoA, which in turn indirectly increases Acetyl-CoA availability to promote de novo lipogenesis. While the effects of changing NME4 expression clearly change lipid content in hepatocytes in vivo and in vitro, the data shown is not sufficient to demonstrate that NME4 increases hepatic lipogenesis. In addition, the logic from the expression data that led to investigate the role of NME4 in hepatocytes is unclear. The major concerns are as follows:

1) The logic of the overlap of NAFLD genes with genes upregulated in adipose tissue is unclear. In the heat map, one can see that NME4 is only upregulated in adipose tissue but not in liver. From this heat map, one would instead investigate the actions of decreasing NME4 expression in adipocytes to see how it affects steatosis in liver by tissue cross-talk. Indeed, UCP1 is not expressed in hepatocytes and alters fatty liver disease by decreasing circulating levels of succinate.

2) The 60% HFD diet used by itself was shown to suppress de novo lipogenesis, namely synthesis of fatty acids from Malonyl-CoA, and mostly promotes steatosis by an increase of the esterification of elevated dietary fatty acids ingested (i.e. endogenous synthesis is suppressed because of excess availability from the diet) (Duarte et al., JLR 2014 and many others). In addition, at 8 and 16 weeks, mitochondrial function and biogenesis can be increased in hepatocytes in early NAFLD, which is recapitulated by a 8 and 16 week high-fat diet feeding in mice (Koliaki et al Cell Met 2015). Therefore, it is a possibility that the increase in NME4 observed in this model is just reflecting an increase in mitochondrial function, in a context where de novo lipogenesis is suppressed.

3) The AAV8 used will also transduce white adipose tissue and knock down NME4 in adipocytes. Therefore, it is a possibility that the effects observed in liver steatosis in vivo are explained by an improvement in white adipose tissue function, to release less fatty acids that would effectively decrease steatosis in liver. Authors should measure the effects of AAV transduction on NME4 expression in BAT, beige and white adipocytes, as well as on the mitochondrial function of these tissues.

4) A Turbo ID assay with the NME4 form that cannot go to the mitochondrial matrix would have been very informative. It is unclear which interaction partners are just reflecting biotinylation mediated by NME4-BirA before the fusion protein is imported to mitochondria. It would have helped to define which interactions might be functionally relevant.

5) It is a possibility that NME4 gain-of-function in hepatocytes promotes steatosis by blocking mitochondrial fatty acid oxidation. In vivo, it could even be possible that NME4 gain of function increased ATP content in mitochondria to block UCP1 activity, inducing white adipose tissue dysfunction to liberate more fatty acids available for the liver, as well as increasing circulating succinate levels. Thus, mitochondrial function oxidizing fatty acids or pyruvate to increase malonyl-coA production must be measured in the models presented.

6) Related to point number 6, no evidence is presented that decreasing the interaction with the enzymes involved in CoA release is sufficient to increase Acetyl-CoA and Malonyl-CoA production. The levels of ATP and other nucleotides in mitochondria are sufficient to determine pyruvate and fatty acid oxidation rates. Therefore, it is still a possibility that a change in the regulation of pyruvate oxidation versus fatty acids oxidation in the mitochondria is the major contributor to the phenotype.

Referee #3:

Xie and coworkers identified NME4 to be important in mediating accumulation of TGs in a mouse model and in human cells. NME4 is localized to the mitochondria and mislocalization abolishes the NME4 mediated effect on lipid accumulation. The data are convincing. I think the authors should rephrase their findings in the discussion. Although the data suggest that NME4 mediates the binding of important lipid metabolic enzymes, it was not shown that this is mechanistically linked with the phenotype. To do so the authors would need to show that the effect would be mediated by an enzymatic inactive NME4 variant. I do not recommend to do the experiment, but to reformulate the discussion.

Review #1:

This is an interesting manuscript that reports the effect of NME4 on NAFLD progression. Xie et al. report that NME4 expression increases in NAFLD and that NME4 inhibition in the liver reduces the susceptibility to High Fat Diet-induced steatosis. Using a remarkable combination of models and methods, the authors suggest that the mechanisms underlying the effect of NME4 on hepatic lipid metabolism involve binding to a significant number of key enzymes in metabolism. While, overall, I think this work represents an impressive amount of work, there are several issues the authors may consider to improve the manuscript:

Thank you for the nice summary and kind suggestions!

1. It seems that NME4 knock-down significantly reduces the expression of lipogenic enzymes in cells. Is it also the case in vivo in the liver of NME4 knock-down? A consistent reduction in the expression of rate-limiting enzymes in fatty acid and TG synthesis could prevent NAFLD. It would also important to consider the possible effect of NME4 on key lipogenic factors such as ChREBP and SREBP1c.

Following the suggestion, we measured the expression levels of key enzymes associated with de novo lipogenesis and triglyceride synthesis in control (AAV-sh*Scramble*) and Nme4 knocked down mice (AAV-sh*Nme4*). Indeed, knocking down Nme4 in mouse hepatocytes significantly decreased the expression of the genes encoding these key enzymes (new Fig 6I), which aligns with the findings observed in cells (Fig EV5H).

Transcription factors ChREBP (Carbohydrate-Responsive Element-Binding Protein) and SREBP1c (Sterol Regulatory Element-Binding Protein 1c) are known to play crucial roles in the regulation of hepatic lipogenesis. Following the suggestion, we assessed the ChREBP and SREBP1c levels using qPCR assay in both cells and mouse tissues. The depletion of NME4 led to a substantial decrease in the expression levels of ChREBP and SREBP1c in both cellular and tissue contexts (new Fig 6F and

EV5G). We added the results to the revised text accordingly (Page 13, line 1-4).

2. I am not sure that NME4 binding to several enzymes involved in metabolic homeostasis is sufficient to modify lipogenesis. This part should be strengthened. One first possible experiment in cell culture would be to measure lipogenic flux in response to change in NME4 modification. In vivo, it would also be interesting to assess the effect of NME4 on the response to a lipogenic challenge (fasting/refeeding high carb) in mice.

Thank you for the nice suggestion! We proceeded to carry out a lipogenic flux assay utilizing a [2-¹³C] malonic acid isotope tracer, in control and Nme4 knocked down mice. Nme4 depletion led to a significant decreased production of malonyl-CoA and acetyl-CoA derived from isotope-labeled malonate in mouse hepatocytes (revised Fig 6F), indicating Nme4 level change indeed impacts the lipogenesis. We have included the results (Page 13, lines 3–6) and a detailed description of the methodology in the revised Methods section (Page 27, lines 21 – Page 28, line 4).

3. Some of the experiments that have been performed with fatty acids (PO) in cell culture. If I understood the work correctly, the concentration of fatty acids used seems very high (1.2 mM).

Thanks for pointing out this issue. Yes, the concentration of fatty acids utilized in our experiments was 1.2 mM, comprising of 0.4 mM palmitic acid and 0.8 mM oleic acid. Prior to conducting these experiments, we reviewed the literature and performed pilot experiments to optimize the condition of the treatment. This concentration has been used in several studies concerning NAFLD. Including “A palmitic acid (PA:0.4 mM; Sigma-Aldrich) and oleic acid (OA:0.8 mM; Sigma-Aldrich) mixture in 0.5% BSA was added to the medium for 24 h to establish an in vitro model of lipid accumulation in hepatocytes.” (Ge *et al*, 2022; Li *et al*, 2021); and “To establish an in vitro model of hepatic steatosis, HepG2 and Huh-7 cells were treated with 1 mM FFA (OA and PA at

a 2:1 vol ratio) in a complete medium containing 1% fatty acid-free bovine serum albumin (BSA) for 24 h.” (Wang *et al*, 2022). We have cited these references in our revised manuscript.

Review #2:

The manuscript of Xie et al characterizes the role of NME in the development of fatty liver disease. They conclude that a maladaptive increase in NME4 expression promotes fatty liver, by NME4 interaction with multiple enzymes that catalyze reactions that release CoA, which in turn indirectly increases Acetyl-CoA availability to promote de novo lipogenesis. While the effects of changing NME4 expression clearly change lipid content in hepatocytes in vivo and in vitro, the data shown is not sufficient to demonstrate that NME4 increases hepatic lipogenesis. In addition, the logic from the expression data that led to investigate the role of NME4 in hepatocytes is unclear. The major concerns are as follows:

Thank you for the careful reading and constructive suggestions! We have extensively revised our manuscript to strengthen the correlation between NME4 and hepatic lipogenesis. We also performed additional experiments to confirm NME4 mainly functions in hepatocytes instead of adipose tissue. We agree that the functional relevance of the interactions between NME4 and the CoA release enzymes has not been fully explored in this study, thus we have tuned down our claims in the revised manuscript.

1. The logic of the overlap of NAFLD genes with genes upregulated in adipose tissue is unclear. In the heat map, one can see that NME4 is only upregulated in adipose tissue but not in liver. From this heat map, one would instead investigate the actions of decreasing NME4 expression in adipocytes to see how it affects steatosis in liver by tissue cross-talk. Indeed, UCP1 is not expressed in hepatocytes and alters fatty liver disease by decreasing circulating levels of succinate.

Thanks for pointing out this issue! Yes, we identified NME4 through our initial bioinformatic analysis of adipose tissue. However, when we were validating our findings and exploring NME4 functions in mouse liver, we found that Nme4 mainly functions in hepatocytes under high-fat diet conditions. Then we realized our heatmap analysis only revealed the NME4 expression patterns in various tissues under normal conditions (original Fig 1B and C).

Thus, we measured Nme4 protein levels in adipose tissues in the normal diet-fed mice and the HFD-fed mice (new Fig EV1G-K and Appendix Fig S1A). Indeed, in healthy adult mice, Nme4 was found to be expressed in both hepatocytes and adipocytes. Nme4 was detected in sWAT and eWAT (new Fig EV1G), as well as in BAT (new Fig EV1H), both in normal diet-fed and in HFD-fed mice (new Appendix Fig S1A). However, there was no significant difference in Nme4 levels in sWAT, eWAT and BAT, between the HFD and normal diet-fed mice (new Fig EV1I-K and Appendix Fig S1A). In contrast, there was a significant increase in Nme4 level in the liver tissue in HFD-fed mice (revised Fig 1I), indicating Nme4 may have specific function in hepatocytes in our HFD-fed mouse model.

To avoid misconceptions, we replaced the dataset utilized in our study with a tissue-based map of the human proteome which identified 2,389 genes highly expressed in liver tissue (Uhlen *et al*, 2015). The NME4 is highly expressed in human liver tissue (revised Fig 1B and 1C, new Fig EV1A). We modified the text accordingly (Page 5, lines 4–11).

2. The 60% HFD diet used by itself was shown to suppress de novo lipogenesis, namely synthesis of fatty acids from Malonyl-CoA, and mostly promotes steatosis by an increase of the esterification of elevated dietary fatty acids ingested (i.e. endogenous synthesis is suppressed because of excess availability from the diet) (Duarte *et al.*, JLR 2014 and many others). In addition, at 8 and 16 weeks, mitochondrial function and biogenesis can be increased in hepatocytes in early NAFLD, which is recapitulated by a 8 and 16 week high-fat diet feeding in mice (Koliaki *et al* Cell Met 2015). Therefore, it is a possibility that the increase in NME4

observed in this model is just reflecting an increase in mitochondrial function, in a context where de novo lipogenesis is suppressed.

Thanks for pointing out this possibility! To test it out, we measured several mitochondrial biogenesis-related gene' expression, including *Ppargc1a*, *Ndufs7*, *Cox5a* and *Cox8b*, in liver tissue from 12- or 24-week normal and HFD-fed mice. The mRNA levels of these genes were slightly decreased in HFD-fed mice, indicating the general mitochondrial function was not increased in our model (new Appendix Fig S1B and C). We have included the result in our revised text (Page 6, lines 1–8).

3. The AAV8 used will also transduce white adipose tissue and knock down NME4 in adipocytes. Therefore, it is a possibility that the effects observed in liver steatosis in vivo are explained by an improvement in white adipose tissue function, to release less fatty acids that would effectively decrease steatosis in liver. Authors should measure the effects of AAV transduction on NME4 expression in BAT, beige and white adipocytes, as well as on the mitochondrial function of these tissues.

Indeed, the AAV8 serotype has been shown to transduce adipose tissue. However, the methods used to target this tissue typically involve orthotopic or intraperitoneal injections in order to maximize efficiency (Jimenez *et al*, 2013; Liu *et al*, 2020). Meanwhile, other studies utilized AAV2/8 for specific gene editing in liver tissue through tail vein injection (Fan *et al*, 2021; Ge *et al.*, 2022). We followed these tail vein injection procedures accordingly to specifically knock down *Nme4* in mouse hepatocytes.

To further assess the impact of AAV2/8 transduction on BAT, sWAT and eWAT in our mouse models, we evaluated the virus adsorption (new Fig 3A and EV3A) as well as the *Nme4* levels (new Fig EV3B) in BAT, sWAT and eWAT in AAV-sh*Nme4* mice. We found the AAV2/8 we used successfully infected liver tissue (new Fig 3A) but not BAT, sWAT or eWAT (new Fig EV3A) (images in Fig 3A and EV3A were taken at the same time with the same settings). The *Nme4* levels remained unaltered in BAT,

sWAT and eWAT, while significantly decreased in liver tissue from *Nme4* knocked down mice (AAV-sh*Nme4*) (new Fig EV3B). Together, these results indicate the AAV2/8-sh*Nme4* virus we used mainly infects mouse liver tissue but not BAT, sWAT or eWAT. We added the results to the revised text (Page 5, line 20 – Page 6, line 1).

4. A Turbo ID assay with the NME4 form that cannot go to the mitochondrial matrix would have been very informative. It is unclear which interaction partners are just reflecting biotinylation mediated by NME4-BirA before the fusion protein is imported to mitochondria. It would have helped to define which interactions might be functionally relevant.

Thanks! Following the suggestion, we constructed a mutant NME4 (mNME4) expression plasmid fused with the TurboID proximity labeling enzyme (new Fig EV4B). As expected, mNME4 lost its ability to localize in the mitochondria (new Fig EV4E). We subsequently performed affinity purification of mNME4 using the TurboID system and processed the results in the same way as the wild-type NME4. In total, we identified 442 high-confidence interacting proteins (HCIPs) (new Appendix Table S10). These HCIPs were characterized using Gene Ontology and Kyoto Encyclopedia of Genes and Genomes analysis (new Appendix Fig S4A and 4B). It was found that these HCIPs mainly localized in the nucleus and cytoplasm and were involved in various biological functions, including the RNA cycle, AMPK autophagy and DNA repair, but not related with mitochondria function or metabolic related processes (new Fig EV4H). Almost all the wild-type NME4 HCIPs were not recovered in mNME4 HCIPs (new Appendix Table S8 and S10), indicating most high-confident NME4 interactors reported were specific. Taken together, these results confirmed the specificity of our NME4 interactome, and highlighted interactions potentially functional relevant. We also revised the text accordingly (Page 10, lines 6–14; Page11, line 14 – Page 12, line 4).

5. It is a possibility that NME4 gain-of-function in hepatocytes promotes steatosis by

blocking mitochondrial fatty acid oxidation. In vivo, it could even be possible that NME4 gain of function increased ATP content in mitochondria to block UCP1 activity, inducing white adipose tissue dysfunction to liberate more fatty acids available for the liver, as well as increasing circulating succinate levels. Thus, mitochondrial function oxidizing fatty acids or pyruvate to increase malonyl-coA production must be measured in the models presented.

Thanks for pointing out this possibility! Following the suggestion, we measured the ATP levels in cells and mice models with NME4 knockout or overexpression. Loss of NME4 has no prominent effect on the level of ATP (new Fig 6G and Figs EV5E and 5F). We added the result to the revised manuscript (Page 12, lines 21-Page 13, line 1). To further understand the details in increased malonyl-CoA production, we performed a lipogenic flux assay utilizing a [2-¹³C] malonic acid isotope tracer, in hepatocytes from control and Nme4 knocked down mice. Nme4 depletion led to a significant decreased production of malonyl-CoA and acetyl-CoA derived from isotope-labeled malonate in mouse hepatocytes (revised Fig 6F). We have included the results (Page 13, lines 3–6) and a detailed description of the methodology in the revised Methods section (Page 27, lines 21 – Page 28, line 4).

6. Related to point number 6, no evidence is presented that decreasing the interaction with the enzymes involved in CoA release is sufficient to increase Acetyl-CoA and Malonyl-CoA production. The levels of ATP and other nucleotides in mitochondria are sufficient to determine pyruvate and fatty acid oxidation rates. Therefore, it is still a possibility that a change in the regulation of pyruvate oxidation versus fatty acids oxidation in the mitochondria is the major contributor to the phenotype.

We agree that the functional relevance of the interactions between NME4 and the CoA release enzymes has not been fully explored in this study, thus we have tuned down our claims in the revised manuscript (Page 15, lines 3–20).

Response to Review #3:

Xie and coworkers identified NME4 to be important in mediating accumulation of TGs in a mouse model and in human cells. NME4 is localized to the mitochondria and mis localization abolishes the NME4 mediated effect on lipid accumulation. The data are convincing. I think the authors should rephrase their findings in the discussion. Although the data suggest that NME4 mediates the binding of important lipid metabolic enzymes, it was not shown that this is mechanistically linked with the phenotype. To do so the authors would need to show that the effect would be mediated by an enzymatic inactive NME4 variant. I do not recommend to do the experiment, but to reformulate the discussion.

Thank you for the positive feedback and nice suggestions. We agree that the functional relevance of the interactions between NME4 and the CoA release enzymes has not been fully explored in this study, thus we have tuned down our claims in the revised manuscript as follows in the revised discussion session (Page 15, lines 3–20). NME4 promotes de novo lipogenesis potentially through the direct binding and activation of key enzymes involved in coenzyme A metabolism, ultimately leading to lipogenesis and steatosis in NAFLD. Among these NME4-binding key enzymes, ACSF3 has been reported to participate in the regulation of fatty acid activation and the synthesis of acetyl-CoA, serving as an important regulatory factor in fatty acid metabolism (Sloan et al, 2011). A previous study has found that the loss of SIRT1 leads to an increase in the level of ACSF3 protein. This increase is believed to be due to the influence of SIRT1 on protein stability, which is regulated by acetylation (Sun et al, 2020). The fluctuation of key enzyme protein levels can lead to disruptions in lipid synthesis and abnormal fatty acid metabolism. Through database analysis, it has been determined that there is a positive correlation between the level of NME4 and CoA metabolism enzymes in liver tissue. This finding suggests that NME4 may play a regulatory role in determining the abundance of these enzymes. Nevertheless, the functional relevance of the interactions between NME4 and the CoA release enzymes

has yet to be explored.

NME4 is an intermediate in histidine phosphorylation. NME4 has the capability to transfer phosphate groups to other proteins (Adam et al, 2020; Fuhs & Hunter, 2017). One hypothesis is that NME4 may phosphorylate these enzymes and subsequently improve protein stability. Further research will be also needed to explore the upstream events of NME4 activation under physiological and pathological conditions.

References

- Adam K, Ning J, Reina J, Hunter T (2020) NME/NM23/NDPK and Histidine Phosphorylation. *International Journal of Molecular Sciences* 21
- Fan L, Lai R, Ma N, Dong Y, Li Y, Wu Q, Qiao J, Lu H, Gong L, Tao Z *et al* (2021) miR-552-3p modulates transcriptional activities of FXR and LXR to ameliorate hepatic glycolipid metabolism disorder. *J Hepatol* 74: 8-19
- Fuhs SR, Hunter T (2017) pHisphorylation: the emergence of histidine phosphorylation as a reversible regulatory modification. *Current Opinion in Cell Biology* 45: 8-16
- Ge C, Tan J, Dai X, Kuang Q, Zhong S, Lai L, Yi C, Sun Y, Luo J, Zhang C *et al* (2022) Hepatocyte phosphatase DUSP22 mitigates NASH-HCC progression by targeting FAK. *Nat Commun* 13: 5945
- Jimenez V, Munoz S, Casana E, Mallol C, Elias I, Jambrina C, Ribera A, Ferre T, Franckhauser S, Bosch F (2013) In vivo adeno-associated viral vector-mediated genetic engineering of white and brown adipose tissue in adult mice. *Diabetes* 62: 4012-4022
- Li YF, Xu JY, Lu YT, Bian H, Yang L, Wu HH, Zhang XW, Zhang BL, Xiong MQ, Chang YF *et al* (2021) DRAK2 aggravates nonalcoholic fatty liver disease progression through SRSF6-associated RNA alternative splicing. *Cell Metabolism* 33: 2004+
- Liu C, Wang J, Wei Y, Zhang W, Geng M, Yuan Y, Chen Y, Sun Y, Chen H, Zhang Y *et al* (2020) Fat-Specific Knockout of Mecp2 Upregulates Slpi to Reduce Obesity by

Enhancing Browning. *Diabetes* 69: 35-47

Sloan JL, Johnston JJ, Manoli I, Chandler RJ, Krause C, Carrillo-Carrasco N, Chandrasekaran SD, Sysol JR, O'Brien K, Hauser NS *et al* (2011) Exome sequencing identifies ACSF3 as a cause of combined malonic and methylmalonic aciduria. *Nat Genet* 43: 883-U896

Sun R, Kang X, Zhao Y, Wang Z, Wang R, Fu R, Li Y, Hu Y, Wang Z, Shan W *et al* (2020) Sirtuin 3-mediated deacetylation of acyl-CoA synthetase family member 3 by protocatechuic acid attenuates non-alcoholic fatty liver disease. *Br J Pharmacol* 177: 4166-4180

Uhlen M, Fagerberg L, Hallstrom BM, Lindskog C, Oksvold P, Mardinoglu A, Sivertsson A, Kampf C, Sjostedt E, Asplund A *et al* (2015) Proteomics. Tissue-based map of the human proteome. *Science* 347: 1260419

Wang Y, Chen CJ, Chen JJ, Sang TT, Peng H, Lin XJ, Zhao Q, Chen SJ, Eling T, Wang XY (2022) Overexpression of NAG-1/GDF15 prevents hepatic steatosis through inhibiting oxidative stress-mediated dsDNA release and AIM2 inflammasome activation. *Redox Biol* 52

Dear Dr. Li,

Thank you for submitting your revised manuscript. It has now been seen by two of the original referees.

As you can see, the referees find that the study is significantly improved during revision and recommends publication. However, I need you to address the points below before I can accept the manuscript.

- Please address the remaining concern of referee #2. Please let me know if you would like to discuss these points further.
- Please provide 3-5 keywords for your study. These will be visible in the html version of the paper and on PubMed and will help increase the discoverability of your work.
- Please rename the 'Availability of data and materials' as 'Data Availability'. Also, please make the datasets publicly available and remove the reviewer passwords from the manuscript.
- Please remove the Author Contributions/CRedit section from the manuscript.
- We note that the Author Checklist is currently missing the information regarding the author name, journal name and the manuscript number (upper left boxes).
- Appendix Tables S1-S4 should be renamed and uploaded as Datasets.
- Appendix Tables S5-S11 should be renamed to EV Tables and be uploaded as such.
- All tables need to have their legends in the corresponding files and the callouts in the manuscript should be updated accordingly
- The Appendix file is currently missing page numbers and a Table of Contents.
- As per our guidelines, please add a 'Data Availability Section', where you state that no data were deposited in a public database.
- We updated our journal's competing interests policy in January 2022 and request authors to consider both actual and perceived competing interests. Please review the policy <https://www.embopress.org/competing-interests> and update your competing interests if necessary. Also, please rename the 'Conflict of Interests' section as 'Disclosure statement and competing interests'.
- The manuscript should follow the following order: Title page - Abstract & Keywords - Introduction - Results - Discussion - Materials & Methods - Data Availability - Acknowledgments - Disclosure Statement & Competing Interests - References - Figure Legends - Tables with legends - Expanded View Figure Legends.
- Appendix Table legends should be removed from the manuscript.
- Our data editors have asked you to clarify the below points in the figure legends:
 - o Please note that a separate 'Data Information' section is required in the legends of figure 3; 7, where the data shown are described (please see an example under <https://www.embopress.org/page/journal/14693178/authorguide#figureformat>)
 - o Please note that legend for figure 7G should start on a new line. OR the legends should be labeled as 'C-G'
 - o Please indicate the statistical test used for data analysis in the legends of figures 1k; 2b-c; 4b, h-i; 5b-c, g-h; EV2d; EV4f; EV5c-d.
 - o Please define the annotated p values * in the legends of figures EV5c-d as appropriate.
 - o Please note that information related to n is missing in the legend of figures 2b; 4b, h-i; EV5c-d.
 - o Please note that the error bars are not defined in the legend of figures EV5c-d.
- Papers published in EMBO Reports include a 'synopsis' and 'bullet points' to further enhance discoverability. Both are displayed on the html version of the paper and are freely accessible to all readers. The synopsis includes a short standfirst summarizing the study in 1 or 2 sentences (max 35 words) that summarize the paper and are provided by the authors and streamlined by the handling editor. I would therefore ask you to include your synopsis blurb and 3-5 bullet points listing the key experimental findings.
- In addition, please provide an image for the synopsis. This image should provide a rapid overview of the question addressed in the study but still needs to be kept fairly modest since the image size cannot exceed 550 pixels wide and 300-600 pixels high.

Thank you again for giving us to consider your manuscript for EMBO Reports, I look forward to your minor revision.

Kind regards,

Deniz Senyilmaz Tiebe

--

Deniz Senyilmaz Tiebe, PhD
Editor
EMBO Reports

Referee #1:

My comments have been addressed thoroughly.
Sorry for a delayed response.

Congratulations for this very nice work.

Referee #2:

The authors successfully addressed most of the comments with the exception of one issue: we still do not know the effects on mitochondrial respiratory function and fatty acid oxidation of NME4 in hepatocytes. The levels of mRNA can be completely uncoupled from mitochondrial protein content and function and thus they are not acceptable measurements of mitochondrial function. Thus, it is still a possibility that NME deletion increases mitochondrial fat oxidation in hepatocytes to explain decreased steatosis and malonyl-coA, while high levels of NME just block fat oxidation. This issue would need to be resolved to define the mechanism of action of NME.

Referee #2:

The authors successfully addressed most of the comments with the exception of one issue: we still do not know the effects on mitochondrial respiratory function and fatty acid oxidation of NME4 in hepatocytes. The levels of mRNA can be completely uncoupled from mitochondrial protein content and function and thus they are not acceptable measurements of mitochondrial function. Thus, it is still a possibility that NME deletion increases mitochondrial fat oxidation in hepatocytes to explain decreased steatosis and malonyl-coA, while high levels of NME just block fat oxidation. This issue would need to be resolved to define the mechanism of action of NME.

Thank you for pointing out this possibility! Since fatty acid β oxidation pathway plays a crucial role in ATP production, we tested whether NME4 depletion changes ATP level in hepatocytes. We found loss of NME4 has no prominent effect on the level of ATP (Fig 6G and Figs EV5E and 5F), indicating NME4 function in hepatocytes is relatively independent to the fatty acid β oxidation pathway.

Following the suggestion, we detected mitochondrial function via Seahorse XFe96 analysis. Cell respiration assays revealed that loss of NME4 didn't significantly increase the mitochondrial function compared with the control group (Fig R1).

Figure R1. Loss of Nme4 in hepatocytes didn't damage mitochondrial function. Seahorse XFe96 analysis of cell respiration in NME4 knockout cells and its corresponding control followed by treatment with OA for 4 hours.

Taken together, we conclude that NME4 does not significantly influence fatty acid oxidation. We agree that the functional relevance of the interactions between NME4 and the CoA release enzymes has yet to be explored, and the mechanism of action of NME4 requires further investigation.

Dear Dr. Li,

Thank you for submitting your revised manuscript. I have now looked at everything and all is fine. Therefore, I am very pleased to accept your manuscript for publication in EMBO Reports.

Congratulations on a nice work!

Kind regards,

Deniz Senyilmaz Tiebe

--

Deniz Senyilmaz Tiebe, PhD

Editor

EMBO Reports

--
